# Multi-objective Bayesian Optimization with Heuristic Objectives for Biomedical and Molecular Data Analysis Workflows

**Alina Selega**                                              *aselega@lunenfeld.ca*
*Lunenfeld-Tanenbaum Research Institute*
*Vector Institute*

**Kieran R. Campbell**                                *kierancampbell@lunenfeld.ca*
*Lunenfeld-Tanenbaum Research Institute*
*University of Toronto*
*Vector Institute*
*Ontario Institute for Cancer Research*

**Reviewed on OpenReview:** *https://openreview.net/forum?id=QspAcsAyis*

## Abstract

Many practical applications require optimization of multiple, computationally expensive, and possibly competing objectives that are well-suited for multi-objective Bayesian optimization (MOBO). However, for many types of biomedical data, measures of data analysis workflow success are often heuristic and therefore it is not known *a priori* which objectives are useful. Thus, MOBO methods that return the full Pareto front may be suboptimal in these cases. Here we propose a novel MOBO method that adaptively updates the scalarization function using properties of the posterior of a multi-output Gaussian process surrogate function. This approach selects useful objectives based on a flexible set of desirable criteria, allowing the functional form of each objective to guide optimization. We demonstrate the qualitative behaviour of our method on toy data and perform proof-of-concept analyses of single-cell RNA sequencing and highly multiplexed imaging datasets for univariate input optimization.

## 1 Introduction

The analysis of high-dimensional biological data is often exploratory and unsupervised. For example, gene expression data may be subject to clustering algorithms to find groups representative of meaningful biological variation. For assays that profile at the patient level, these clusters may represent novel disease subtypes, while for assays at the single-cell level, they may represent novel cell types.

Despite the importance of these methods, there is no "one-size-fits-all" approach to the analysis of such data. Instead, there is a myriad of different possible parameter combinations that govern these workflows and lead to variations in the results and interpretation. For example, in the analysis of single-cell RNA-sequencing (scRNA-seq) – a technology that quantifies the expression profile of all genes at single-cell resolution – a common analysis strategy is to cluster the cells to identify groups with biological significance. However, each workflow for doing so has variations with respect to data normalization, cell filtering strategies, and the choice of clustering algorithm and parameters thereof. Changes to these algorithm and parameter choices produce dramatically different results (Germain et al., 2020; Duò et al., 2018) and there is no ground truth available. This motivates an important question: how do we optimize these workflows such that the resulting exploratory analysis best reflects the underlying biology? It is important to note the presence of measurement noise in virtually all biomedical data, which can arise from the technology used for data acquisition or represent underlying biological heterogeneity.

In the adjacent field of supervised machine learning (ML), such optimization over workflows has largely been tackled from the perspective of automated ML (AutoML, He et al. (2021)). This comprises a diverse set of methods such as Bayesian optimization (Snoek et al., 2012) and Neural Architecture Search (Elsken et al.,

2018) that attempt to optimize the success of the model with respect to one or more hyperparameter settings. In this context, success is defined as the model accuracy on a held out test set, though can also correspond to the marginal likelihood of the data given the model and hyperparameters.

However, in the context of exploratory analysis of genomic data, existing AutoML approaches face three challenges. Firstly, they are almost exclusively unsupervised, meaning there is no notion of accuracy on a test set we may optimize with respect to. Secondly, the majority of methods are not generative probabilistic models (Zappia et al., 2018) so it is impossible to optimize with respect to the marginal or test likelihood. Finally, the objectives used to optimize a workflow are numerous, conflicting, noisy, due to the underlying noise present in the raw data, and can be highly subjective, due to often being heuristics.

This is demonstrated by attempts to benchmark clustering workflows of scRNA-seq data. As said above, there are many parameters that must be set, e.g. which subset of genes and clustering algorithm to use, along with such parameters as resolution in the case of community detection (Germain et al., 2020). However, there is no quantitative way to choose which parameter setting is "best" and so the community turns to a number of heuristic objectives to quantify the performance of a workflow. For example, Cui et al. (2021) attempt to optimize the adjusted Rand index (ARI) with respect to expert annotations and a heuristic based around downsampling rare cell types while minimizing runtime. Germain et al. (2020) similarly consider the ARI but also the average silhouette width to maximize cluster purity. Zhang et al. (2019) consider a range of heuristics including agreement with simulated data and robustness to model misspecification.

However, given that these objectives are all heuristic and open to user preference, there is no guarantee that all of them are *useful* and have maxima that align with the meta-objective at hand, which in the above example is the ability to identify a biologically relevant population of cells. Conversely, some heuristic objectives may be *non-useful* – they are largely noisy and attribute nothing to the overall optimization problem by not aligning with a meta-objective. For example, in an Imaging Mass Cytometry experiment, which also aims to cluster cells, an antibody that quantifies the expression of a given protein may fail entirely, which would not be identified prior to data analysis. In that case, any objective that included that protein's expression would be irrelevant to the meta-objective, but this would not be known up front. This motivates the central question we attempt to address: how can we adapt AutoML approaches to optimize unsupervised workflows over multiple heuristic objectives that are frequently subjective and conflicting?

To begin to tackle this question, we introduce MANATEE (Multi-objective bAyesiaN optimizAtion wiTh hEuristic objEctives). The key idea is that by considering a linear scalarization as a probabilistic weighting over (heuristic) objective inclusion, we may up- or downweight an objective based on desirable or non-desirable properties of its posterior functional form. Consequently, rather than returning the full Pareto front that may include points (parameter values) that maximize potentially non-useful heuristic objectives, we automatically concentrate on a useful region in accordance with the specified properties. In this work, we evaluate our method only for univariate input optimization; while we designed our framework to be extensible to multivariate inputs, we do not make any claims with regards to performance in that setting here. The main contributions we present are:

1. Introduce the concept of *behaviours* $\mathcal{B}$ of the posterior functional form of the surrogate objective function $\mathbf{f}$ that are desirable if a function is *useful* for overall optimization.

2. Suggest an example set of such behaviours that may be inferred from the posterior of a multi-output Gaussian process, if used as the surrogate function.

3. Build upon previous MOBO approaches with random scalarizations that compute the distribution of scalarization weights $p(\boldsymbol{\lambda})$ but instead condition on objective behaviours with $p(\boldsymbol{\lambda}|\mathcal{B})$, inferring which objectives are useful.

4. Construct a set of example objectives measuring workflow success for real molecular imaging and transcriptomic data and show that the proposed procedure compares favourably to existing approaches for univariate input optimization.

## 2 Background

### 2.1 Bayesian optimization

Bayesian optimization (BO, see Frazier (2018) and references therein) attempts to optimize a function $g(x) \in \mathbb{R}$ for some $x \in \mathbb{R}^D$ that is, in some sense, expensive to evaluate and for which derivative information is not available, precluding gradient-based approaches. Note that while we set the scene in the general case of a multivariate input, in this work we perform experiments with univariate inputs, though our approach can be extended to $D > 1$ in the future. Applications of BO have become popular in the tuning of ML hyperparameters (Turner et al., 2021) and indeed entire workflows (Fusi et al., 2018) due to the expensive nature of re-training the models.

At their core, BO approaches propose a surrogate function $f$ defined on the same range and domain as $g$ that may be searched efficiently to find points $x$ that either maximize $g$, reduce uncertainty about $f$, or both. This leads to the concept of an acquisition function[1] $\mathrm{acq}(x) \in \mathbb{R}$ that may be optimized to find the next $x$ at which $g$ may be evaluated. While multiple acquisition functions have been proposed, here we focus on the Upper Confidence Bound (UCB) (Auer, 2002) defined as:

$$\mathrm{acq}_{\mathrm{UCB}}(x) = \mu^{(t)}(x) + \sqrt{\beta_t}\sigma^{(t)}(x) \tag{1}$$

where $\mu^{(t)}(x)$ and $\sigma^{(t)}(x)$ are the posterior mean and standard deviation of $f$ at $x$ after $t$ acquisitions from $g$, while $\beta_t$ is a hyperparameter that controls the balance between exploration and exploitation. While there are many possible choices for the surrogate function $f$, including deep neural networks (Snoek et al., 2015), a popular choice is a Gaussian process due to its principled handling of uncertainty and capacity to approximate a wide range of functions.

### 2.2 Gaussian processes

**Overview** Gaussian processes (GPs) (Williams & Rasmussen, 2006) define a framework for performing inference over nonparametric functions. Let $m(x)$ be a mean function and $k(x, x')$ a positive-definite covariance function for $x, x' \in \mathbb{R}^D$. We define $f(x)$ to be a Gaussian process denoted $f(x) \sim \mathcal{GP}\left(m(x), k(x, x')\right)$ if for any finite-dimensional subset $\mathbf{x} = [x_1, \dots, x_N]^T \in \mathbb{R}^{N \times D}$, the corresponding function outputs $\mathbf{f} = [f(x_1), \dots, f(x_N)]$ follow a multivariate Gaussian distribution $p(\mathbf{f}|\mathbf{x}) = \mathcal{N}(\mathbf{0}, \mathbf{K})$, where $\mathbf{K}$ is the covariance matrix with entries $(\mathbf{K})_{ij} = k(x_i, x_j)$ and we have assumed a zero-mean function without loss of generality. The kernel fully specifies the prior over functions, with one popular choice we use throughout the paper being the *exponentiated quadratic* kernel $k(x, x') = \exp\left(\frac{(x-x')^2}{l^2}\right)$. It is common to model noisy observations $\mathbf{y}$ via the likelihood $p(\mathbf{y}|\mathbf{f})$, which when taken to be $\mathcal{N}(\mathbf{f}, \sigma_\epsilon^2)$ with noise variance $\sigma_\epsilon^2$ leads to the exact marginalization of $\mathbf{f}$.

**Multi-output Gaussian processes** GPs may be extended to model $K$ distinct outputs[2] via the functions $\{f_k(x)\}_{k=1}^K$ (Bonilla et al., 2007). One construction is to model the full covariance matrix as the Kronecker product between the $K \times K$ inter-objective covariance matrix $\mathbf{K}^{\mathrm{IO}}$ and the data covariance matrix:

$$\mathrm{cov}\left(f_k(x), f_{k'}(x')\right) = (\mathbf{K}^{\mathrm{IO}})_{k,k'} k(x, x'). \tag{2}$$

Here the kernel hyperparameter $l$ is shared across objectives, though in the following we model objective-specific observation noises $\epsilon_k$ with variances $\sigma_{\epsilon_k}^2$ as $p(y_k|f_k) \sim N(f_k, \sigma_{\epsilon_k}^2)$.

### 2.3 Multi-objective Bayesian optimization

**Multi-objective optimization** Multi-objective optimization attempts to simultaneously optimize $K$ objectives $g_1(x), \dots, g_K(x)$ over $x \in \mathbb{R}^D$, which is common in many real-world settings (Deb, 2014). However, it is rare in practice to be able to optimize all $K$ functions simultaneously and instead is common to attempt to recover the *Pareto front*. We say a point $x_1$ is *Pareto dominated* by $x_2$ iff $g_k(x_1) \leq g_k(x_2) \, \forall k = 1, \dots, K$

---

[1]We will later refer to it as the "single-objective acquisition function".
[2]Commonly referred to as *tasks*, we here refer to them as *objectives* given the application.

and $\exists\ k \in 1, \ldots, K$ s.t. $g_k(x_1) < g_k(x_2)$. A point is said to be *Pareto optimal* if it is not dominated by any other point. The *Pareto front* is then defined as the set of Pareto optimal points, which intuitively corresponds to the set of equivalently optimal points given no prior preference between objectives.

**Scalarization functions** One popular approach to multi-objective optimization is the use of *scalarization functions* (see Chugh (2020) for an overview). A scalarization function $s_{\boldsymbol{\lambda}}(\mathbf{g}(x))$ parameterized by $\boldsymbol{\lambda}$ takes the set of $K$ functions $\mathbf{g}(x) = [g_1(x), \ldots, g_K(x)]$ and outputs a single scalar value to be optimized in lieu of $\mathbf{g}(x)$. Roijers et al. (2013) show that if $s_{\boldsymbol{\lambda}}$ is monotonically increasing in all $g_k(x)$ then the resulting optimum $x^*$ lies on the Pareto front of $\mathbf{g}$.

While many scalarization functions exist, one popular choice is the linear scalarization function $s_{\boldsymbol{\lambda}}(\mathbf{g}(x)) = \sum_{k=1}^{K} \lambda_k g_k(x),\ \lambda_k > 0\ \forall k$. This has the intuitive interpretation that each $\lambda_k$ corresponds to the weight of function $k$, with a larger relative value pulling the optimum of $s_{\boldsymbol{\lambda}}$ towards the optimum of $g_k$.

**Hypervolume improvement** Another multi-objective optimization approach relies on the notion of *hypervolume* (HV), the volume of the space dominated by a Pareto front and bounded from below by a reference point, which current work assumes to be known by the practitioner (Daulton et al., 2021). HV is used as a metric to assess the quality of a Pareto front and is sought to be maximized in the optimization. Expected HV improvement (EHVI) for a new set of points can be computed using box decomposition algorithms (Yang et al., 2019a).

**Multi-objective Bayesian optimization with scalarizations** Multi-objective Bayesian optimization (MOBO) approaches that use scalarizations operate under the same conditions as BO, where each evaluation of $g_k(x)$ is expensive and derivative information is unavailable. An example method is ParEGO (Knowles, 2006), which randomly scalarizes objectives with augmented Chebyshev scalarization and uses expected improvement. It was recently extended to $q$NParEGO (Daulton et al., 2020), which supports parallel and constrained optimization in a noisy setting. Unlike hypervolume-based methods which can struggle with $> 5$ objectives (Balandat et al., 2021), $q$NParEGO is more suited for such problems.

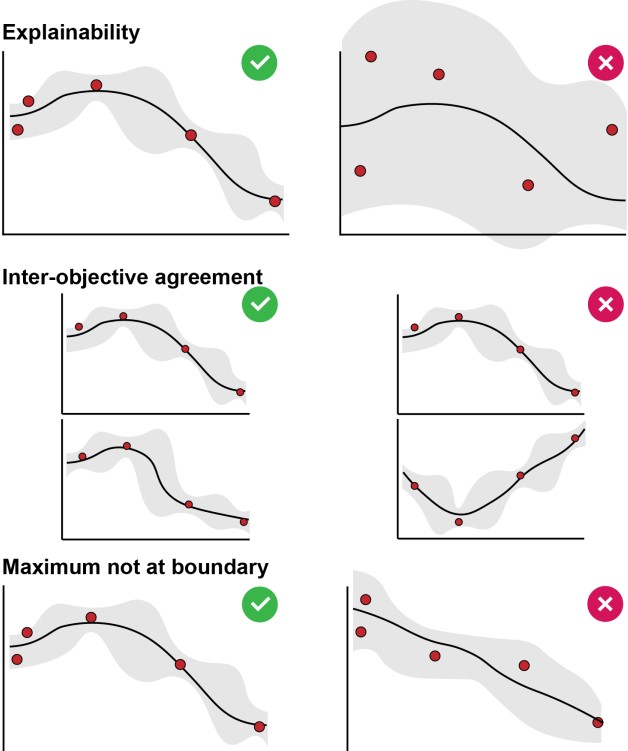

Figure 1: Cartoon illustrating proposed desirable behaviours of objectives. *Explainability* captures how much an objective covaries with the parameter $x$, favouring those with lower observation noise. *Inter-objective agreement* favours objectives that agree with each other. *Maximum not at boundary* determines whether the optimum is contained within the user-specified parameter range.

Paria et al. (2019) propose a MOBO procedure that, rather than maximizing $s_{\boldsymbol{\lambda}}$ for a single $\boldsymbol{\lambda}$, constructs a distribution $p(\boldsymbol{\lambda})$ and minimizes the expected pointwise regret,

$$\mathcal{R}(\mathbf{X}) = \mathbb{E}_{p(\boldsymbol{\lambda})} \left( \max_{x \in \mathcal{X}} s_{\boldsymbol{\lambda}}(\mathbf{g}(x)) - \max_{x \in \mathbf{X}} s_{\boldsymbol{\lambda}}(\mathbf{g}(x)) \right),$$

where $\mathcal{X}$ is the feature space of $x$ and $\mathbf{X}$ is the subset of $\mathcal{X}$ lying on the Pareto front to be computed. The exact region of the Pareto front to be considered is governed by $p(\boldsymbol{\lambda})$ and the authors provide a bounding box procedure for the user to select $p(\boldsymbol{\lambda})$, akin in the case of a linear scalarization to asserting *a priori* which objectives $k$ are important. However, to our knowledge, no MOBO approach has proposed a $p(\boldsymbol{\lambda}|\cdot)$, inferred

from either the data or the posterior over functions, that adaptively up- or downweights objectives based on desirable properties.

**Multi-objective Bayesian optimization beyond scalarizations**  For hypervolume-based methods in the MOBO setting, EHVI has been extended to parallel evaluation of $q$ points, leveraging automatic differentiation and boosting efficiency (Daulton et al., 2020). As EHVI assumes noise-free case and can be affected in noisy settings, recent work introduced noisy EHVI (NEHVI), which uses its expectation under the posterior distribution of the surrogate function values given noisy observations (Daulton et al., 2021). NEHVI is more robust to noise than other hypervolume-based MOBO methods, is equivalent to EHVI in the noiseless setting, and its parallel formulation ($q$NEHVI) achieves computational gains and state-of-the-art performance in large batch optimization (Daulton et al., 2021).

Another approach to MOBO utilizes uncertainty, overcoming the issue of scalability with the number objectives faced by hypervolume-based methods. Predictive entropy search for multi-objective optimization (PESMO) aims to minimize entropy of the posterior distribution over the Pareto set (Hernández-Lobato et al., 2016). Max-value Entropy Search for Multi-objective Optimization (MESMO) employs efficient output space entropy search, improving its computation time over PESMO (Belakaria et al., 2019). Uncertainty-aware Search framework for optimizing Multiple Objectives (USeMO) selects points that maximize a multi-objective measure of uncertainty and outperforms existing methods on problems with up to six objectives with faster convergence (Belakaria et al., 2020).

### 2.4   Applications of AutoML and Bayesian optimization in molecular biology and genomics

AutoML and BO approaches have previously been successfully applied across multiple problems in molecular biology and genomics. One example is a popular application of BO to generate protein candidates at the sequence level with desirable chemical properties (Yang et al., 2019b). While less common, there are a handful of examples of AutoML applications to hyperparameter optimization in genomics. The GenoML project (Makarious et al., 2021) provides a Python framework centered on open science principles to perform end-to-end AutoML procedures for supervised learning problems in genomics. AutoGeneS (Aliee & Theis, 2021) develops a multi-objective optimization framework for the selection of genes for the deconvolution of bulk RNA-sequencing without relying on marker genes and instead optimizing properties of identified cell clusters. However, to our knowledge, there is no work that tackles the general problem of optimizing bioinformatics and genomics workflows in the absence of well-defined objective functions. In contrast, there are multiple BO techniques that allow a user to express a preference between solutions (González et al., 2017). While these methods could have exciting applications in genomics, we consider an alternative setup, where the user expresses a preference on the functional form of the unseen objectives but does not participate during acquisition.

## 3   Multi-objective Bayesian optimization over heuristic objectives

### 3.1   Setup

We assume we have access to $K$ noisy, heuristic objectives that at acquisition step $t$ return a measurement $y_{kt}$ for an input location $x_t \in \mathcal{X}$, where $\mathcal{X}$ is a compact subset of $\mathbb{R}$ on $[a, b]$. We introduce surrogate functions $f_k(x)$ that we model with a multi-output GP as described in Section 2.2 with a full kernel given by Eq. 2. The choice to fit a multi-output GP to data reflects our prior assumption that the heuristic objectives may have a correlated functional form or operate on similar lengthscales. In settings where these assumptions do not hold, using a multi-output GP may be suboptimal. Our framework is applicable to any scalarization function that uses weightings (e.g. Tchebyshev scalarization (Nakayama et al., 2009)) and here we consider a linear scalarization function over objectives $s_{\boldsymbol{\lambda}}(\mathbf{f}(x)) = \sum_k \lambda_k f_k(x)$. Ultimately, we seek to maximize $\mathbb{E}_{p(\boldsymbol{\lambda}|\cdot)}[s_{\boldsymbol{\lambda}}(\mathbf{f}(x))]$.

### 3.1.1 Acquisition functions

The next point to query $x_{t+1}$ is chosen by maximizing the expectation of the acquisition function. For this we propose two approaches:

- SA (scalarized acquisition): maximize the expectation of the scalarization of the single-objective acquisition function of each objective $\mathbb{E}_{p(\boldsymbol{\lambda}|\cdot)}\left[s_{\boldsymbol{\lambda}}(\text{acq}(\mathbf{f}(x)))\right]$ as per Paria et al. (2019);

- AS (acquisition of scalarized): maximize the expectation of the single-objective acquisition function of the scalarized objectives $\mathbb{E}_{p(\boldsymbol{\lambda}|\cdot)}\left[\text{acq}(s_{\boldsymbol{\lambda}}(\mathbf{f}(x)))\right]$.

We refer to these as acquisition functions and derive expressions for both in Appendix C. While many choices of single-objective acquisition functions are possible, we use $\text{acq}_{\text{UCB}}$ (Eq. 1), the UCB single-objective acquisition function. The SA formulation simplifies to an intuitive interpretation where each objective's UCB function value is weighted by the probability of that objective being useful given its behaviours (Appendix C). The AS formulation takes into account the multi-objective posterior covariance structure (Appendix C) but has a longer computation time that may require approximations when $K$ is large.

### 3.2 Desirable heuristic objective behaviours

Next, we wish to set $p(\boldsymbol{\lambda}|\cdot)$ to upweight objectives that are inferred as useful based on desirable properties learned from the data. In our framework, we assume that the practitioner has *a priori* no preference over objectives, only that some or all may be useful. Instead, they have a preference over the functional form of the objectives, which is expressed via some desirable properties. We begin by considering what properties of a given heuristic objective $f_k(x)$ may be considered desirable. While many are possible, we suggest three *behaviours* (Figure 1):

1. **Explainability:** $f_k(x)$ covaries significantly with $x$ (i.e. is explained by $x$). The justification here is that the practitioner has selected heuristic $k$ assuming it will provide insight into the choice of $x$, so if there is no correlation then it should be downweighted. Given that the data have been scaled to empirical variance 1, the inferred variance of the fitted observation noise $\sigma_{\epsilon_k}^2$ represents the proportion of variance unexplained by $f_k$ so we define $B_k^{(1)} := \sigma_{\epsilon_k}^2$.

2. **Inter-objective agreement:** $f_k$ shares a similar functional form with $f_{k'}, k' \neq k$, with the intuition that it is useful for practitioners to find regions of the input space where multiple heuristics agree. After fitting the multi-output GP, $(\mathbf{K}^{\text{IO}})_{k,k'}$ defines the covariance between objective $k$ and $k'$ for $k \neq k'$ and $(\mathbf{K}^{\text{IO}})_{k,k}$ defines the variance of objective $k$. We therefore introduce the inter-objective agreement behaviour as

$$B_k^{(2)} := \sum_{k'=1 \,:\, k' \neq k}^{K} \max\left(0, \frac{1}{K-1}\frac{(\mathbf{K}^{\text{IO}})_{k,k'}}{\sqrt{(\mathbf{K}^{\text{IO}})_{k,k}(\mathbf{K}^{\text{IO}})_{k',k'}}}\right). \tag{3}$$

The intuition is that $\frac{(\mathbf{K}^{\text{IO}})_{k,k'}}{\sqrt{(\mathbf{K}^{\text{IO}})_{k,k}(\mathbf{K}^{\text{IO}})_{k',k'}}}$ represents the correlation between objectives $k$ and $k'$ so $B_k^{(2)}$ represents the average correlation with other objectives while not penalizing negative correlation worse than no correlation.

3. **Maximum not at boundary:** Within $\mathcal{X}$, $f_k$ has a maximum that is not at the boundary of $x$. The useful range of $x$ is specified by the practitioner. Then, if $f_k$ is maximized by a boundary value of $x$, then either (i) the optimum is outside of the specified range, conflicting with the practitioner's intuition, or (ii) $f_k$ is unbounded in $x$, in which case it is not useful for optimization. In the former case, one approach would be to revise the range and repeat the process. Since the derivative of a GP is also a GP, we may identify whether a stationary point exists in $\mathcal{X}$ by searching for the zeros of the posterior mean derivative $\bar{f}'(x)$. We therefore define $B_k^{(3)} := \text{hasmax}(f_k, \mathcal{X})$, where hasmax returns 1 if $f_k$ has a maximum on $\mathcal{X}$ and 0 otherwise by evaluating the derivatives of the posterior mean of the multi-output GP (derived in Appendix D).

We will denote a set of three above behaviours for a given objective $k$ as $\mathbf{B}_k = \{B_k^{(1)}, B_k^{(2)}, B_k^{(3)}\}$.

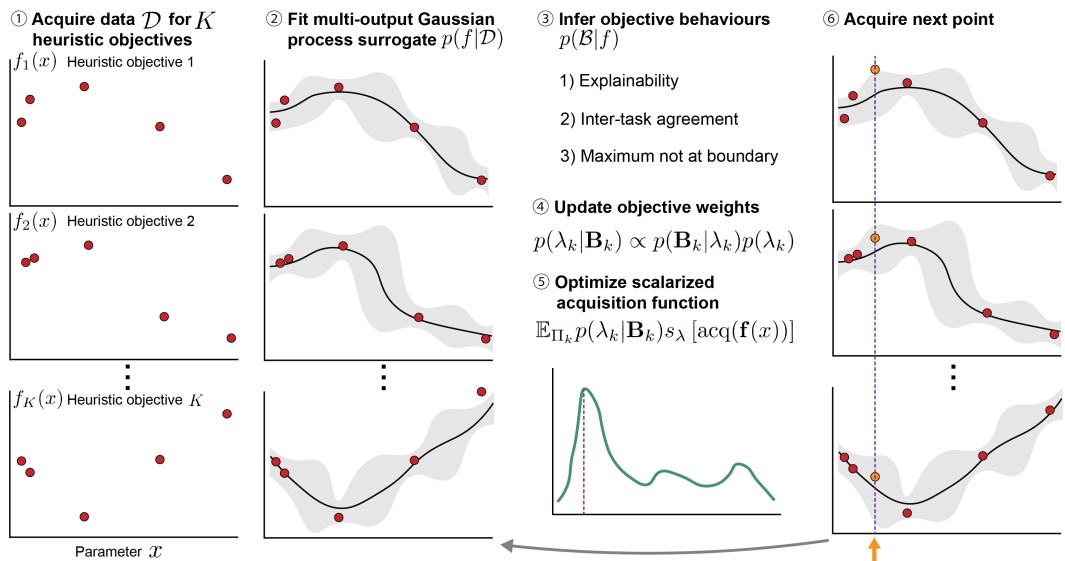

Figure 2: Multi-objective Bayesian optimization with heuristic objectives. A multi-output Gaussian process is fitted (Step 2) to the initial training dataset (Step 1). Objective behaviours are inferred from the posterior form (Step 3) and are used to update the distribution of objective inclusion weights (Step 4). The acquisition function defined as the expectation under the weights distribution is optimized (Step 5) to give the next location to sample objectives at (Step 6). Step 5 shows the scalarized acquisition (SA) function approach.

### 3.3 Incorporating desirable behaviours into scalarization weights

We next consider how to use the set of behaviours $\mathcal{B} = \{\mathbf{B}_k\}_{k=1}^{K}$ to parameterize the scalarization probabilities $p(\boldsymbol{\lambda}|\mathcal{B})$. We assume that $\lambda_k$ is a binary variable $\forall k$ that corresponds to whether objective $k$ is *useful* or otherwise, with $p(\lambda_k|\mathbf{B}_k)$ given by a Bernoulli distribution. While this construction initially appears restrictive compared to existing approaches with random scalarizations where $\boldsymbol{\lambda} \in \mathbb{R}^K$ , it has a desirable property that maintains its generality (note that we assume $s_{\boldsymbol{\lambda}}$ to be a linear scalarization function). Specifically, optimization under our setup returns Pareto optimal points of $\mathbf{f}$ (proof presented in Appendix E).

**Theorem 3.1.** *If* $\mathbb{E}_{p(\lambda_k|\mathbf{B}_k)}[\lambda_k] > 0\ \forall k,\ then\ x^* = \arg\max_x \mathbb{E}_{p(\boldsymbol{\lambda}|\mathcal{B})}[s_{\boldsymbol{\lambda}}(\mathbf{f}(x))]\ lies\ on\ the\ Pareto\ front\ of\ \mathbf{f}.$

However, how to construct $p(\lambda_k = 1|B_k^{(1)}, B_k^{(2)}, B_k^{(3)})$ directly is non-obvious. Instead, we ask how would *each* objective behaviour appear if we knew that objective was useful (or otherwise)? This allows us to specify conditional distributions over behaviours for a useful/non-useful objective $p(B_k^{(i)}|\lambda_k = 1)$, $p(B_k^{(i)}|\lambda_k = 0)$ for $i = 1, 2, 3$ and combine with a prior $p(\lambda_k = 1) = 1 - p(\lambda_k = 0)$ to compute the posterior probability that an objective is useful given its behaviours:

$$p(\lambda_k = 1|\mathbf{B}_k) = \frac{\prod_i p(B_k^{(i)}|\lambda_k = 1)p(\lambda_k = 1)}{\sum_{q=0,1} \prod_i p(B_k^{(i)}|\lambda_k = q)p(\lambda_k = q)} \tag{4}$$

With these considerations, we suggest distributions for $p(B_k^{(i)}|\lambda_k)$; however, we emphasize that these are suggestions only and there are many possible that would fit the problem.

**Explainability:** For $B_k^{(1)}$, the explainability of objective $k$ (i.e. the proportion of variance unexplained by that function), we assume that if that objective is desirable ($\lambda_k = 1$) then the lower the observation noise, the better and in the non-desirable case ($\lambda_k = 0$), higher noise is expected. Given the lack of additional assumptions, we appeal to the principle of parsimony and propose a linear relationship of the form:

$$p(B_k^{(1)}|\lambda_k) = \begin{cases} 2(1 - \lambda_k)B_k^{(1)} + 2\lambda_k(1 - B_k^{(1)}) & \text{if } B_k^{(1)} \in [0, 1] \\ 0 & \text{otherwise.} \end{cases} \tag{5}$$

**Inter-objective agreement:**  For inter-objective agreement $B_k^{(2)}$, our reasoning is that identifying solutions where multiple objectives agree can be desirable for a practitioner. We thus posit that high inter-objective correlation should be more likely for a desirable objective and vice-versa for a non-desirable one, and again a linear relationship is the most parsimonious:

$$p(B_k^{(2)}|\lambda_k) = \begin{cases} 2\lambda_k B_k^{(2)} + 2(1-\lambda_k)(1-B_k^{(2)}) & \text{if } B_k^{(2)} \in [0,1] \\ 0 & \text{otherwise.} \end{cases} \tag{6}$$

**Maximum not at boundary:**  We propose $B_k^{(3)}|\lambda_k = i \sim \text{Bernoulli}(\pi_i)$ where $\pi_0, \pi_1$ are user-settable hyper-parameters. This means that conditioned on an objective being useful (or otherwise), there is a fixed probability of that objective containing a maximum in the region. In our experiments, we set $\pi_0, \pi_1$ such that a useful objective has a maximum in the region with 75% chance, and a non-useful one with 25% chance.

$$p(B_k^{(3)}|\lambda_k) = \begin{cases} \text{Bernoulli}(\pi_0) & \text{if } \lambda_k = 0 \\ \text{Bernoulli}(\pi_1) & \text{if } \lambda_k = 1 \end{cases} \tag{7}$$

### 3.4  MANATEE

Putting these steps together results in the MANATEE framework, an iterative MOBO procedure as outlined in Figure 2 and Algorithm 1. First, the objectives are evaluated at a set of input locations randomly chosen on the parameter space. Second, the multi-output GP surrogate function with covariance given by Eq. 2 is fitted to all objectives. Then, the objective behaviours $\mathcal{B}$ are computed from the surrogate function and the distributions over objective weights that indicate whether each objective is useful are updated. Finally, the updated acquisition function is optimized (see Section 3.1.1 and Appendix C), guiding acquisition of the next point. The procedure is repeated for a predetermined number of steps. The overall "best" point to be used for downstream analysis may be chosen as that which maximizes the scalarized surrogate function.

---

**Algorithm 1** MANATEE framework

---

**Input:** training data $\mathcal{D}^{(0)}$; input space $\mathcal{X}$ on region $[a,b]$; objectives $\mathbf{f}(x) = \{f_k(x)\}_{k=1}^K$; acquisition function $\mathcal{A}(\mathbf{f}(x), s_{\boldsymbol{\lambda}}, \text{acq})$; behaviours $\mathbf{B} = \{B^{(1)}, B^{(2)}, B^{(3)}\}$; distributions over behaviours for useful/non-useful objective $p(B_k^{(i)}|\lambda_k)$; number of iterations $T$

**for** $t \leftarrow 1, \ldots, T$ **do**
    Fit multi-output Gaussian process $p(f|\mathcal{D}^{(t-1)})$
    **for** $k \leftarrow 1, \ldots, K$ **do**
        **for** $i \leftarrow 1, 2, 3$ **do**
            Compute objective behaviour $B_k^{(i)}$ from the Gaussian process posterior $f$ (Section 3.2)
            Compute $p(B_k^{(i)}|\lambda_k)$ (how likely $B_k^{(i)}$ is given objective is useful/non-useful) using Eq. 5, 6, 7
        **end for**
        Update $p(\lambda_k = 1|\mathbf{B}_k)$ (probability that objective is useful given its behaviours) using Eq. 4
    **end for**
    Update acquisition function $\mathcal{A}(\mathbf{f}(x), s_{\boldsymbol{\lambda}}, \text{acq})$ using objective weights $p(\boldsymbol{\lambda}|\mathcal{B})$ (Section 3.1.1)
    Pick next candidate by maximizing acquisition function $x_t \leftarrow \arg\max_{x \in \mathcal{X}} \mathcal{A}(\mathbf{f}(x), s_{\boldsymbol{\lambda}}, \text{acq})$
    Acquire data at new location $y_k(x_t) \; \forall k = 1, \ldots, K$ objectives
    Update training data with new datapoints $\mathcal{D}^{(t)} \leftarrow \mathcal{D}^{(t-1)} \cup \{(x_t, \mathbf{y}(x_t))\}$
**end for**

---

### 3.5  Baselines for experiments

We contrast our method against two baselines and three existing approaches for MOBO: (i) *Random acquisition*: draw $x_t \sim \text{Unif}(0,1)$ at each iteration, (ii) *Random scalarization*: use identical surrogate and acquisition functions as MANATEE-SA to sample $x_t$ but draw $\lambda_k \sim \text{Unif}(0,1)$ rather than conditional on $\mathcal{B}$, (iii) $q$NEHVI (Daulton et al., 2021) with approximate hypervolume computation to facilitate inference over

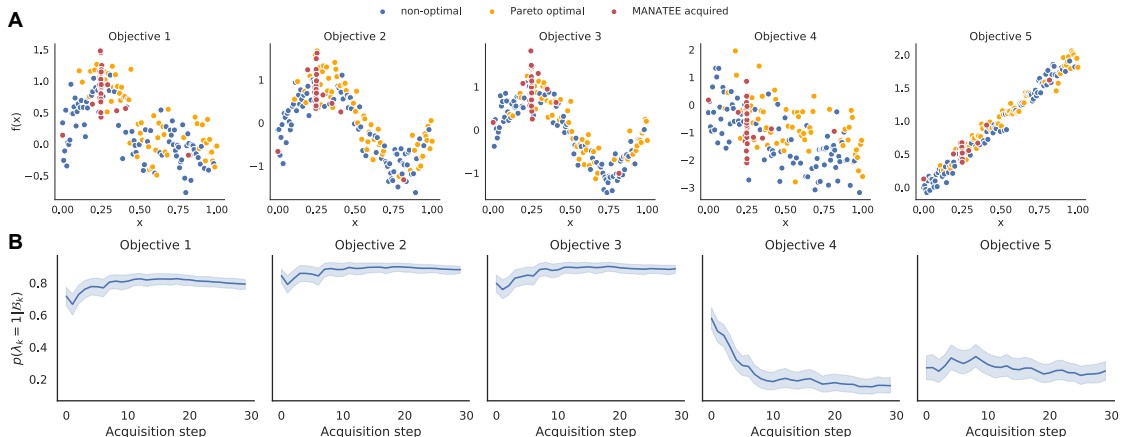

Figure 3: **A** 150 random samples of toy data for the 5 objectives, including those on the Pareto front (orange) and otherwise (blue), along with points acquired by MANATEE-SA (red) in an example run. **B** Inclusion probabilities for each of the objectives as a function of acquisition step. Solid line shows the mean and shaded region denotes the 95% confidence interval across runs.

$> 5$ objectives, (iv) $q$NParEGO (Daulton et al., 2020), and (v) USeMO (Belakaria et al., 2020). Since it is not known how noisy real biomedical problems are, we supplement our evaluation including $q$NEHVI and $q$NParEGO, specifically designed for MOBO of noisy objectives, with USeMO, an efficient uncertainty-based state-of-the-art approach for many objectives in the noiseless setting (Belakaria et al., 2020).

In our setup not all objectives are considered useful, so quantifying the hypervolume improvement over all objectives as a commonly used performance measure (Daulton et al., 2020; 2021) does not align with the stated goal. Thus, we construct evaluation measures (referred to as *meta-objectives*) that use independent information (e.g. expert labels, see Sections 4.2, 4.3) to quantify the quality of the acquisitions.

When a meta-objective $h(x_t)$ is available at every iteration $t = 1, \ldots, T$ with overall maximum $y^* = \max_{x \in \mathcal{X}} h(x)$, to compare among approaches we compute the following metrics: (i) *Cumulative regret:* $\frac{1}{T} \sum_{t=1}^{T} (y^* - h(x_t))$, (ii) *Full regret:* $y^* - \max_{x \in X_{1:T}} h(x)$, and (iii) *Bayes regret:* $\frac{1}{T} \sum_{t=1}^{T} (y^* - \max_{x \in X_{1:t}} h(x))$, where $X_{1:t}$ is the set of $x$ acquired up to time $t$. Of these, we place most emphasis on cumulative regret as it quantifies how close each method gets to the optimal solution on average. In contrast, the full and Bayes regret quantify how close the "best" acquired point gets to $y^*$ as measured by the max over $h$ of all points acquired so far; however, since the meta-objective $h$ is in general inaccessible for our problem setup (and only used for method comparison), it is impossible to quantify $\max_{x \in X_{1:T}} h(x)$ in practice outside of benchmarking exercises.

## 4 Experiments

### 4.1 Toy data experiment

We begin by demonstrating the overall problem setup on toy data on an input space $x \in [0, 1]$. We consider 5 objectives overall – 3 that act as the *useful* objectives with maxima around that of a meta-objective at $1/4$ given by $\sin 2\pi x$, $\max(0, \sin 2\pi x)$, and $\sin 2\pi(x - 0.05)$ and 2 that disagree and act as the *non-useful* objectives given by $2x$ and $-2x$. Each objective is augmented with noise (Appendix B.1). Note that on real data we do not know *a priori* which objectives are useful[3]. Further, the meta-objective is not specified – it may be linear, non-linear, and not necessarily a function of the heuristic objectives – it simply needs a maximum at $x \approx \frac{1}{4}$.

---

[3]Otherwise only useful objectives would be included and standard MOBO procedures applied.

Samples from each of these functions can be seen in Figure 3A (blue points). The overall Pareto front (orange points) spans almost the entire region including samples at the very right where one of the non-useful linear objective functions has its maximum. However, when applied to this toy problem, MANATEE quickly begins acquiring samples around the joint maxima of the three useful objective functions (red points). Indeed, tracing the inclusion probabilities $p(\lambda_k = 1|\mathbf{B}_k)$ across the iterations (Figure 3B) demonstrates how MANATEE learns to upweight objectives 1-3 while downweighting 4-5. This demonstrates than when we do not know *a priori* which objectives to trust, we may still recover a region of high utility when the Pareto front spans the full space of conflicting objectives.

## 4.2 Imaging Mass Cytometry cofactor selection

We next apply MANATEE to the selection of cofactors for Imaging Mass Cytometry (IMC) data, a new technology that can measure the expression of up to 40 proteins at subcellular resolution in tissue sections (Giesen et al., 2014). In the analysis of mass cytometry data, a cofactor $c$ is frequently used to normalize the data (Ray & Pyne, 2012; Wagner et al., 2019) via the transformation $\tilde{y} = \sinh^{-1}(y/c)$. However, to our knowledge no systematic approach exists to set the cofactor and it is typically left as a user-specified parameter.

Table 1: Results for IMC cofactor optimization experiment. CR: cumulative regret, FR: full regret; BR: Bayes regret. M-SA: MANATEE with scalarized acquisition, M-AS: MANATEE with acquisition of scalarized function, RA: random acquisition, RS: random scalarization. ARI: adjusted Rand index, NMI: normalized mutual information. Values are mean (s.d.).

| Method | ARI | | | NMI | | |
|---|---|---|---|---|---|---|
| | CR | FR | BR | CR | FR | BR |
| M-SA | **0.017(0.005)** | 0.003(0.003) | **0.007(0.006)** | **0.019(0.009)** | **0.002(0.005)** | **0.008(0.009)** |
| M-AS | 0.021(0.010) | 0.006(0.010) | 0.011(0.011) | 0.025(0.017) | 0.007(0.016) | 0.015(0.017) |
| RA | 0.045(0.001) | 0.024(0.013) | 0.031(0.008) | 0.065(0.003) | 0.026(0.013) | 0.036(0.009) |
| RS | 0.021(0.004) | 0.003(0.004) | 0.008(0.005) | 0.025(0.006) | 0.003(0.006) | **0.008(0.007)** |
| qNEHVI | 0.042(0.004) | 0.011(0.007) | 0.021(0.011) | 0.061(0.007) | 0.011(0.007) | 0.026(0.017) |
| qNParEGO | 0.037(0.005) | **0.002(0.003)** | 0.017(0.012) | 0.049(0.009) | 0.005(0.003) | 0.023(0.017) |
| USeMO | 0.043(0.003) | 0.010(0.009) | 0.016(0.009) | 0.062(0.005) | 0.012(0.011) | 0.018(0.013) |

Here, we consider the standard workflow where (i) the expression data is normalized with a given cofactor $c$ and (ii) the data is clustered using standard methods with the "best" cofactor being the one that leads to the most biologically relevant cellular populations[4]. Given that this problem in general has no notion of "test accuracy" with respect to which we could optimize the cofactor, we instead suggest a number of heuristic objectives based around maximizing the correlation of cluster-specific mean expression of known protein marker combinations. For example, the proteins CD19 and CD20 are highly expressed in B lymphocyte cells and lowly expressed in all others. Therefore, if a clustering correctly separates B cells from others, the correlation between the mean CD19 and CD20 expression in each cluster should be high as the proteins should either be co-expressed or both not expressed (at the origin), as demonstrated in Appendix F.1. We can apply this logic to a range of cell type markers to construct our set of heuristic objectives (Appendix F.2).

To quantify the ability of each clustering to uncover biologically relevant populations, we use expert annotated cell types from Jackson et al. (2020) and assess cluster overlap with the adjusted Rand index (ARI) and normalized mutual information (NMI), which for this experiment form the overall meta-objectives [5] in line with prior benchmarking efforts of single-cell clustering (Qi et al., 2020; Kiselev et al., 2017). Note that this is in general unavailable for the analysis of newly generated data and we would *only* have access to the correlation (heuristic) objectives.

---

[4]All parameters of the clustering procedure are held constant across cofactors to allow for fair comparison.
[5]The fact that we can easily specify 2 meta-objectives highlights the ubiquity of the "multiple heuristic objective" issue in bioinformatics.

The results comparing MANATEE to the alternative methods are shown in Table 1 and the optimization trajectories across acquisitions are shown in Supplementary Figure 4. On the metric of cumulative regret, which as above, is most relevant for the problem setup at hand, MANATEE-SA outperforms the alternative approaches. On full and Bayes regret, MANATEE performs comparably with the baselines. On cumulative regret, $q$NEHVI and USeMO are comparable to random acquisition, suggesting that consistently acquiring close-to-optimal solutions over $> 5$ noisy objectives is challenging even for approximate hypervolume computation and that a noisy setting is challenging for methods assuming noise-free observations. Interestingly, we find that random scalarization exhibits strong performance on several measures, which may be understood by the fact that the scalarized objective $\sum_k \lambda_k f_k$ naturally places high weight on regions where many objectives agree, mimicking a similar scenario to our inter-objective agreement criterion. Samples from the objectives along with methods' acquisitions are shown in Supplementary Figure 5. MANATEE samples low cofactor values corresponding to regions where meta-objectives' values are high, while other methods sample throughout the parameter range. Examining the inclusion probabilities shows that MANATEE learns to downweight the CD45/CD20 co-expression objective, which is maximized for high cofactor values (Supplementary Figures 5 and 6). We also show each regret metric as a function of the acquisition step to demonstrate the convergence rate of each method (Appendix A.5, Supplementary Figure 10a). MANATEE-SA reaches low regret values faster than or comparable to other methods.

We further performed ablation experiments of each behaviour and found that no single behaviour drives the performance (Appendix A.4). We also performed cross-validation on data splits to demonstrate that MANATEE does not overfit to a given dataset (Appendix A.3).

## 4.3   Single-cell RNA-seq highly variable gene selection

Single-cell RNA-sequencing (scRNA-seq, see Hwang et al. (2018) for an overview) quantifies whole-transcriptome gene expression at single-cell resolution. A key step in the analysis of the resulting data is selection of a set of highly variable genes (HVGs) for downstream analysis, typically taken as the "top $x\%$" (Yip et al., 2019), but there are no systematic or quantitative recommendations for selecting this proportion (Luecken & Theis, 2019). Therefore, we apply MANATEE to this problem following a clustering workflow similar to the IMC experiment, but by varying the proportion of HVGs used for the analysis and keeping all other clustering parameters fixed. We again propose a number of co-expression based heuristics (Appendix F.3) and augment these with measures of cluster purity (mean silhouette width, Calinski and Harabasz score, Davies-Bouldin score) previously used in scRNA-seq analysis (Germain et al., 2020).

Table 2: Results for scRNA-seq HVG selection optimization experiment. CR: cumulative regret, FR: full regret; BR: Bayes regret. M-SA: MANATEE with scalarized acquisition, M-AS: MANATEE with acquisition of scalarized function, RA: random acquisition, RS: random scalarization. ARI: adjusted Rand index, NMI: normalized mutual information. Values are mean (s.d.).

| Method | ARI | | | NMI | | |
|---|---|---|---|---|---|---|
| | CR | FR | BR | CR | FR | BR |
| M-SA | **0.126(0.019)** | 0.056(0.010) | 0.064(0.012) | 0.125(0.026) | 0.022(0.010) | 0.031(0.013) |
| M-AS | 0.127(0.026) | 0.058(0.015) | 0.067(0.015) | **0.124(0.036)** | 0.023(0.013) | 0.034(0.017) |
| RA | 0.192(0.020) | 0.053(0.009) | 0.067(0.013) | 0.199(0.025) | 0.025(0.009) | 0.040(0.015) |
| RS | 0.140(0.018) | **0.049(0.008)** | **0.059(0.010)** | 0.130(0.021) | **0.014(0.007)** | **0.026(0.012)** |
| qNEHVI | 0.186(0.038) | 0.050(0.008) | 0.092(0.045) | 0.191(0.047) | 0.023(0.009) | 0.073(0.057) |
| qNParEGO | 0.161(0.039) | 0.055(0.009) | 0.091(0.048) | 0.158(0.049) | 0.023(0.009) | 0.070(0.064) |
| USeMO | 0.218(0.031) | 0.057(0.013) | 0.071(0.014) | 0.231(0.040) | 0.027(0.014) | 0.042(0.019) |

For these workflows, no general ground truth clustering or cell types are available. However, a new technology called CITE-seq can simultaneously quantify both the RNA and surface protein expression at single-cell level (Stoeckius et al., 2017). Given that cell types are traditionally defined by surface protein expression (Oostrum et al., 2019), we use a clustering of the surface protein expression alone as the ground truth following existing work (Liu et al., 2021). The concordance with this clustering acts as the meta-objective

in this experiment, which we benchmark the proposed approaches against. We supply each method with the heuristic objectives above and benchmark the gene proportion acquisitions by contrasting the resulting clusterings with the surface protein-derived ground truth using ARI and NMI as metrics. Once again, these represent only two possible choices of meta-objective and there are many more we could design, highlighting the prevalence of heuristic objectives in the field.

The results are shown in Table 2, the optimization trajectories in Supplementary Figure 7, and regret metrics (Appendix A.5) at each acquisition step in Supplementary Figure 10b. As above, our main focus is on cumulative regret since when deploying in a real-world scenario, we would not have access to the meta-objective. MANATEE performs favourably on cumulative regret compared to the other approaches, though has higher full and Bayes regret. MANATEE learns to strongly upweight the Davies-Bouldin objective, which agrees with the meta-objectives and is less noisy, and downweight the CD3E/CD4 and CD3D/CD8A objectives, which are maximized in a different region to the meta-objectives (Supplementary Figures 8 and 9). This allows our method to acquire proportions of HVGs close to the upper bound of the parameter range where the meta-objectives are maximized, unlike $q$NParEGO and USeMO that don't reach this region (Supplementary Figure 8). Overall, this demonstrates our method to be a promising approach to tackle hyperparameter optimization on real, noisy datasets and achieve competitive performance compared to existing baselines and state-of-the-art methods.

Finally, we quantified the effects of different model initializations on the parameter values selected by our method by reanalyzing the outputs of the above experiments, which contained multiple repetitions across different random seeds, controlling the initial training set and model initialization. In the IMC cofactor selection experiment, MANATEE-SA and MANATEE-AS most frequently selected the same cofactor value in 99% and 90% of all repetitions (Supplementary Figure 11a). In the HVGs selection experiment, the interquartile range of percentage values most frequently selected by MANATEE-SA in all repetitions was 5% and that for MANATEE-AS was 8%, while the considered range of optimization was between 1% and 50% of genes (Supplementary Figure 11b). All implementation details are provided in Appendix A.1 and the experimental setup is described in Appendix A.2.

## 5 Discussion

A common theme here is the subjectivity of parameter setting in biological data analysis workflows. Setting these often involves no heuristic objectives at all, simply relying on an iterative data exploration to find a parameter combination that "works". Even when heuristic objectives are involved – such as in the benchmarking analyses of scRNA-seq workflows – the precise choice of which objectives to include is fundamentally subjective too.

It is important to note that our proposed approach does not remove subjectivity from the analysis. Many important steps, including the chosen behaviours $\mathcal{B}$ and their conditional inclusion distributions $p(\mathcal{B}|\boldsymbol{\lambda})$ are set by the user. Therefore, it abstracts the subjectivity by a level, changing the question from *"which objectives should I use to benchmark my method?"* to *"what would the behaviour of a good objective function be?"*. Given that no link is assumed between the specified heuristic objectives and the true meta-objective and that the choice of desirable objective behaviours is given as example only, we make no optimality claims about the ability to explore the Pareto front.

In our setup, weighting of the acquisition functions may effectively exclude some objectives and lead to potential under-exploration. This may be mitigated by starting the objective weighting process after some set number of steps. Furthermore, the inter-objective agreement behaviour may be potentially limiting if a practitioner is interested in jointly suboptimal solutions that still perform reasonably well on all objectives, some of which are competing. If all objectives had the maximum at the boundary, this behavior would no longer be relevant to the optimization; in such cases, the user could be alerted and asked to re-define which behaviours they want to use. As each behaviour's weight depends on the specification of $p(\mathcal{B}|\boldsymbol{\lambda})$, there is no guarantee that these would have a similar scale, though we found no one behaviour driving the performance on the problems we considered here. Here we have assumed that all tasks are quantifiable for all observations, though in some Bayesian optimization settings this doesn't hold (Krause & Ong, 2011).

Our framework can be used with other scalarization functions that include a weighting $\boldsymbol{\lambda}$ which may be more suitable in some scenarios, but here we performed experiments using linear scalarization. We note that if the true meta-objective is not linear in the observed objectives then a linear scalarization will be suboptimal. However, in the context we considered, the true meta-objective may represent complex notions such as the ability to uncover biologically meaningful results, which, while likely nonlinear, is impossible to quantify or specify. Therefore, the linear scalarization represents a trade-off between having access to an intuitive weight $\lambda_k$ for each function while not necessarily being theoretically optimal. In addition, MANATEE performed competitively with state-of-the-art methods in our experiments with real data analysis pipelines, which represent cases with unknown true meta-objective.

As our general approach is applicable to any multi-objective optimization scenario (while tailored to biomedical analyses), we acknowledge that it could be used in highly diverse applications. We note that these may include ethically dubious bioinformatics analyses such as those pertaining to genetic testing of embryos. We strongly caution against any such use without a thorough ethical review process.

There are several extensions that would serve as future steps. We have only considered optimizing $x \in \mathbb{R}$, but future work can use our method to optimize multiple pipeline parameters as our work generalizes to $x \in \mathbb{R}^D$ for $D > 1$ (Appendix D). There is much current research in BO methods over both continuous and categorical domains (Ru et al., 2020), which may better suit the parameter space of scRNA-seq analysis pipelines (Germain et al., 2020). A lot of research in BO centres on the incorporation of user input and expert opinions to guide optimization (Häse et al., 2021; Abdolshah et al., 2019). While we have explicitly considered the opposite problem – where *a priori* it is not known which objectives should be upweighted – there could be situations where both approaches could be integrated. For example, an expert may provide ratings for the results of each scRNA-seq clustering during optimization. In such settings, these ratings could be integrated into our proposed framework by updating the distributions $p(\boldsymbol{\lambda}|\mathcal{B}, \Theta)$ over $\Theta$ such that they confer high weights to functions of expert ratings. Finally, we welcome future work evaluating gains in discovery and accuracy of biological results and computation time arising from using our method to choose values for pipeline parameters.

## Code availability

Our code is available at `https://github.com/camlab-bioml/2022_manatee_paper_analyses`.

## Acknowledgments

We thank Dr. Gavia Gray for her advice on PyTorch optimizers and the reviewers for their thoughtful comments and suggestions. This work was supported by a Canadian Institutes of Health Research project grant number PJT175270 to K.R.C, the Canadian Statistical Sciences Institute Ontario Top-up for Postdoctoral Fellows in Data Science to A.S., the Hold'em for Life Oncology Fellowship to A.S., and the Vector Institute Postgraduate Affiliate Award to A.S. We acknowledge the support of the Natural Sciences and Engineering Research Council of Canada (NSERC) RGPIN-2020-04083 to K.R.C. This research was undertaken, in part, thanks to funding from the Canada Research Chairs Program to K.R.C.

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

# A    Implementation details

## A.1    Method hyperparameters

Hyperparameters for all methods are summarized in Supplementary Table 3.

**MANATEE**    MANATEE (with both AS and SA acquisition functions) was implemented with PyTorch v. 1.9.0 (Paszke et al., 2019) and `gpytorch` v. 1.6.0 (Gardner et al., 2018) for the Gaussian process model and inference. Optimization was performed with the LBFGS optimizer. At every acquisition step, the model was initialized and fit to to the current training set 5 times and the model with the highest log-likelihood was kept. If fitting failed, the process would be re-tried a maximum of 20 times before halting. Optimization of the acquisition function was initialized with the maximum of 100 random samples. Above implementation details also apply to the random scalarization (RS) baseline. Optimization of the SA and AS acquisition functions was performed with the LBFGS optimizer with line search. For MANATEE, maxima of the posterior mean were identified by computing the first derivative of the posterior mean, finding its zeros using Brent's method (Brent, 2013) implemented by `scipy.optimize.brentq` with default parameters, and computing the second derivative at those locations. A candidate was declared a maximum not at boundary if its second derivative was less than -10 and if the candidate was at least 0.01 units away from the range extrema.

**qNEHVI**    The *q*NEHVI approach was implemented with `botorch` v. 0.6.1.dev37+g4f0a2889 (Balandat et al., 2020). The development version was used to facilitate usage of *q*NEHVI with the `KroneckerMultiTaskGP` model (Maddox et al., 2021). Implementation closely followed the tutorial on multi-objective Bayesian optimization (Balandat et al., 2021). Batch size was set to 1. `fit_gpytorch_model` was called with `max_retries` set to 20. Other parameters were set following the tutorial. Reference point was set to the minimum of the initial acquired points. To accommodate > 5 objectives, we used approximate hypervolume

Supplementary Table 3: Method hyperparameters used in the experiments. RS: random scalarization.

| Method | Parameter | Value | Explanation |
|---|---|---|---|
| MANATEE | $p(\lambda_k = 1)$ | 0.5 | Prior over binary $\lambda_k$ is set as Bernoulli(0.5) |
| | $\pi_1, \pi_0$ | 0.75, 0.25 | Bernoulli hyperparameters for $p(B_k^{(3)}|\lambda_k=i)$ |
| | $\frac{\delta^2 f}{\delta x^2} <$ | -10 | Upper bound to call a max |
| | Min distance | 0.01 | Distance from max to extrema |
| | $l >$ | 0.1 | Kernel lengthscale constraint |
| | $\sigma^2$ | 1 | Kernel variance |
| | $\sigma^2_{\epsilon_k} >$ | 0.01 | Observation noise variance constraint |
| | Line search function | `strong_wolfe` | LBFGs optimizer arg |
| MANATEE RS | UCB $\beta_t$ | $0.125 \log(2t + 1)$ | Set as in Paria et al. (2019) |
| | GP fits | 5 | Model inits and fits at each acquisition |
| | GP fit re-tries | 20 | Max re-tries to fit model at each acquisition |
| | Acquisition samples | 100 | Initial samples from acquisition function |
| $q$NEHVI $q$NParEGO | `MC_SAMPLES` | 128 | QMC sampler arg, set as in Balandat et al. (2021) |
| | `max_retries` | 20 | `botorch.fit_gpytorch_model` arg |
| | `batch_range` | $(0, -1)$ | QMC sampler arg, set as in Balandat et al. (2021) |
| | GP model | `KroneckerMultiTaskGP` | Set as in Balandat et al. (2021) |
| | `BATCH_SIZE` | 1 | Points to acquire |
| | `NUM_RESTARTS` | 20 | Optimization re-starts, set as in Balandat et al. (2021) |
| | `RAW_SAMPLES` | 1024 | Acquisition samples, set as in Balandat et al. (2021) |
| | `batch_limit` | 5 | `optimize_acqf` arg, set as in Balandat et al. (2021) |
| | `maxiter` | 200 | `optimize_acqf` arg, set as in Balandat et al. (2021) |
| $q$NEHVI | reference point | sample minimum | Lower HV bound |
| | `alpha` | $10^{-8+\#objectives}$ | approximate partitioning level |
| USeMO | reference point | $10^5$ | Set as in `github.com/belakaria/USeMO` |
| | acquisition function | TS | Thompson sampling (Thompson, 1933) |
| | `batch_size` | 1 | Points to acquire |
| | d | 1 | Input dimensionality |
| | `beta` | $\log\left(\frac{t^{\frac{d}{2}+2}\pi^2}{0.15}\right)$ | Set as in `github.com/belakaria/USeMO` |
| | algorithm | NSGA-II Deb et al. (2002) | Set as in `github.com/belakaria/USeMO` |
| | function evalutions | 2500 | Set as in `github.com/belakaria/USeMO` |

computation by setting the `alpha` parameter according to the heuristic based on the number of objectives as proposed in (Daulton et al., 2020).

**$q$NParEGO**  The $q$NParEGO approach was implemented with `botorch` v. 0.6.1.dev37+g4f0a2889 (Balandat et al., 2020). Implementation closely followed the tutorial (Balandat et al., 2021). Batch size was set to 1. `fit_gpytorch_model` was called with `max_retries` set to 20. Other parameters were set following the tutorial.

**USeMO** USeMO was implemented using the code deposited in the `http://github.com/belakaria/USeMO` repository, with `main.py` adapted to work with pipelines for the IMC and scRNA-seq experiments. USeMO was run with the Thompson sampling (TS) acquisition function (Thompson, 1933) as USeMO-TS and USeMO-EI (EI, expected improvement (Mockus et al., 1978)) were shown to outperform existing methods (Belakaria et al., 2020) and TS didn't require a hyperparameter to set like the exploration/exploitation trade-off hyperparameter in the implementation of the EI acquisition function. All other hyperparameters were left as in `main.py`.

### A.2 Experimental setup

Supplementary Table 4: Parameters of the experimental procedure.

| Experiment | Parameter | Value |
|---|---|---|
| Toy | Number of initial points | 5 |
| | Number of acquisitions | 30 |
| | Replicates | 100 |
| | $x_{min}$ | 0 |
| | $x_{max}$ | 1 |
| | Number of objectives | 5 |
| IMC | Number of initial points | 5 |
| | Number of acquisitions | 35 |
| | Replicates | 98 |
| | $x_{min}$ | 1 |
| | $x_{max}$ | 100 |
| | Number of objectives | 7 |
| scRNA-seq | Number of initial points | 5 |
| | Number of acquisitions | 36 |
| | Replicates | 100 |
| | $x_{min}$ | 0.01 |
| | $x_{max}$ | 0.5 |
| | Number of objectives | 9 |

Parameters for all experimental procedures are summarized in Supplementary Table 4.

**Toy experiment** The Pareto front on toy objectives was computed with the OApackage (Eendebak & Vazquez, 2019). The initial dataset contained 5 training points at random locations and MANATEE-SA performed 30 acquisition steps. The experiment was repeated 100 times. The range of the optimized parameter $x$ was between 0 and 1. Toy objectives are described in Appendix B.1.

**IMC experiment** At each acquisition step, data was normalized with the acquired cofactor value and clustered. Mean marker expression in each cluster was computed on the data normalized with cofactor=1. The co-expression objective values were computed as a Pearson correlation between the mean expression of a marker pair across clusters. The experiment was repeated 98 times. The range of the optimized cofactor value was between 1 and 100. The overall meta-objective maximum for ARI and NMI, used to compute regrets, was set as the maximum of ARI/NMI values computed from all acquisitions by all methods (MANATEE-SA, MANATEE-AS, RA and RS baselines, $q$NEHVI, $q$NParEGO, USeMO), including behaviour ablation methods (MANATEE-SA with leave-one-out behaviour). $q$NEHVI returned an error and completed fewer than the total of 35 acquisitions on 49 runs; $q$NParEGO returned an error and completed fewer than the total of 35 acquisitions on 80 runs, other methods completed all acquisitions on all runs. The objectives are described in Appendix F.2.

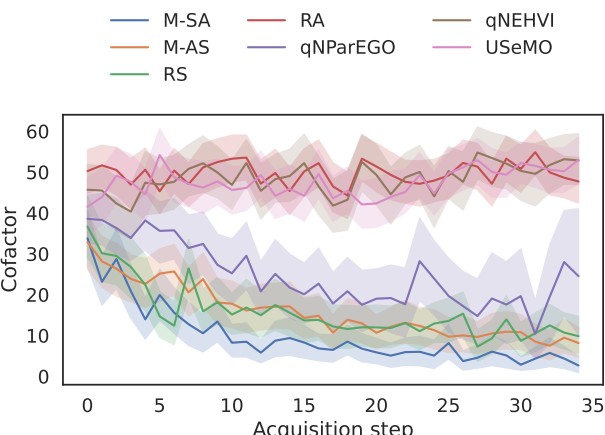

Supplementary Figure 4: Optimized cofactor value as a function of acquisition step for all methods. M-SA: MANATEE with scalarized acquisition, M-AS: MANATEE with acquisition of scalarized function, RA: random acquisition, RS: random scalarization. Solid line shows the mean and shaded region denotes the 95% confidence interval.

**scRNA-seq experiment**  At each acquisition step, data was subsetted to the top highly variable genes according to the acquired proportion value and clustered. The co-expression objective values were computed as a Pearson correlation between the mean expression of a marker pair across clusters. The experiment was repeated 100 times. The range of the optimized highly variable gene proportion was between 0.01 and 0.5. Unsupervised cluster purity metrics were computed on the PCA transform of the normalized scRNA-seq data. The overall meta-objective maximum for ARI and NMI, used to compute regrets, was set as the maximum of ARI/NMI values computed from all acquisitions by all methods (MANATEE-SA, MANATEE-AS, RA and RS baselines, $q$NEHVI, $q$NParEGO, USeMO), including behaviour ablation methods (MANATEE-SA with leave-one-out behaviour). $q$NEHVI returned an error and completed fewer than the total of 36 acquisitions on 64 runs, $q$NParEGO returned an error and completed fewer than the total of 36 acquisitions on 62 runs, other methods completed all acquisitions on all runs. The objectives are described in Appendix F.3.

Experimental results were tracked with Weights & Biases (Biewald, 2020). Reported regrets include acquisitions from incomplete runs for $q$NEHVI and $q$NParEGO.

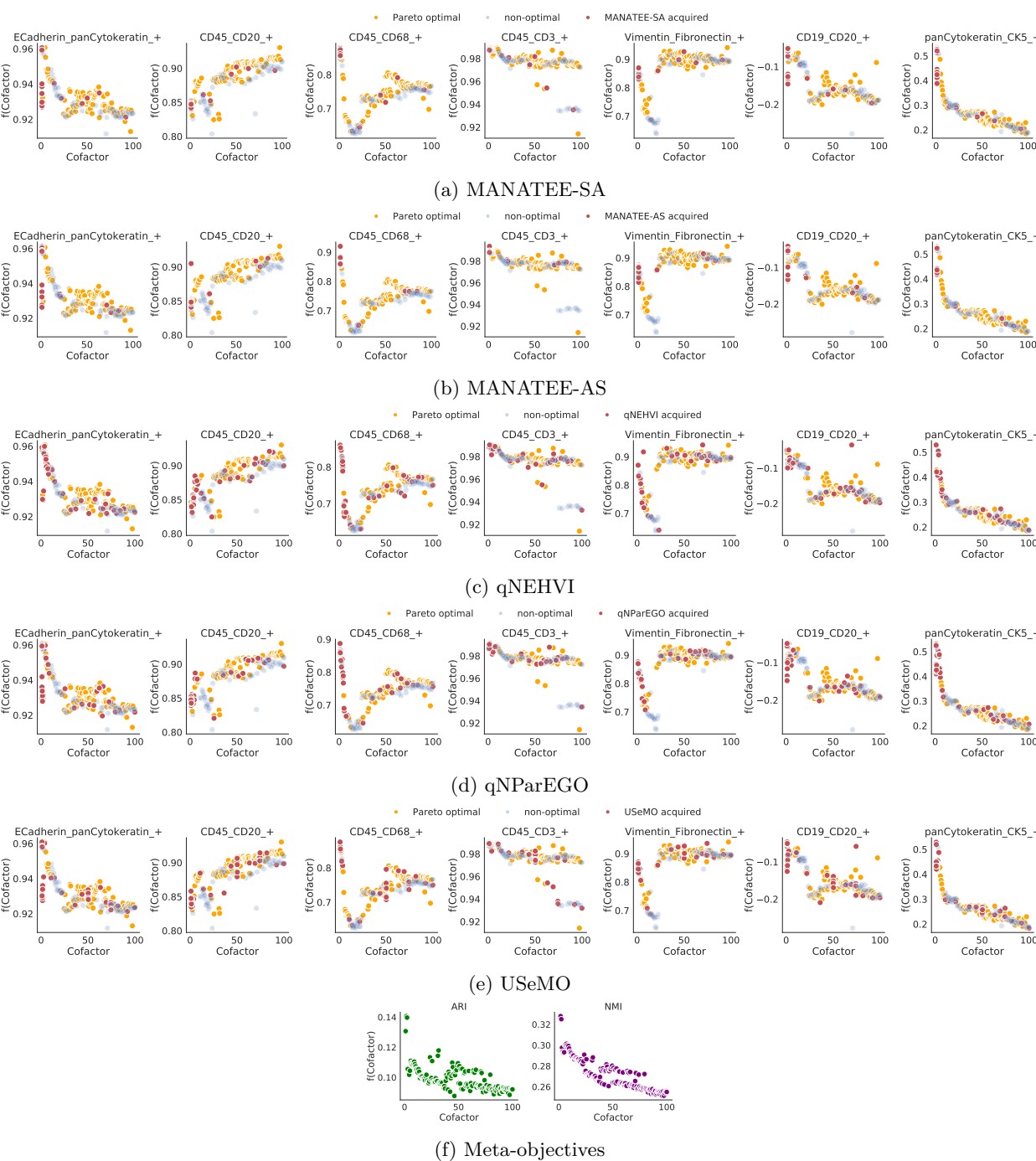

Supplementary Figure 5: 150 random samples from 7 co-expression objectives and meta-objectives (ARI, NMI) in the IMC cofactor selection experiment. Orange indicates samples on the Pareto front, blue indicates Pareto dominated samples, and red indicates points acquired by each method. The acquisitions shown for each method are from the run with the highest average ARI value. Pareto optimality of random samples was computed with the OApackage (Eendebak & Vazquez, 2019).

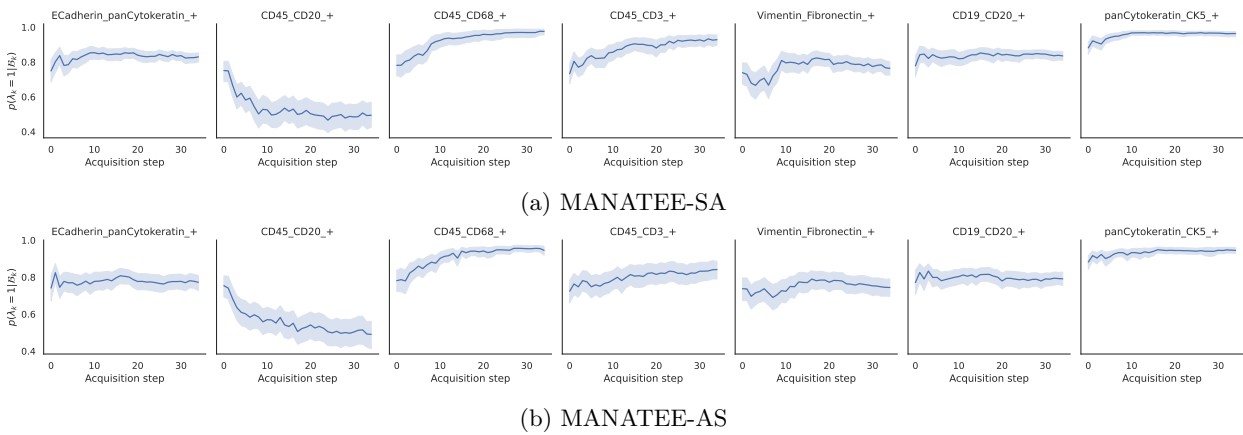

(a) MANATEE-SA

(b) MANATEE-AS

Supplementary Figure 6: Inclusion probabilities for MANATEE-SA and MANATEE-AS for each of the objectives as a function of acquisition step in the IMC cofactor selection experiment. Solid line shows the mean and shaded region denotes the 95% confidence interval across runs.

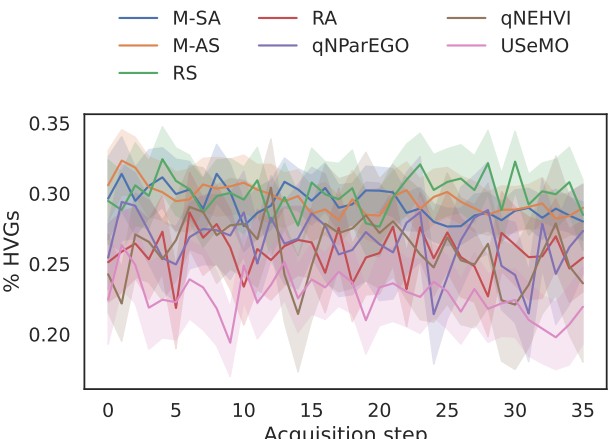

Supplementary Figure 7: Optimized percentage of highly variable genes as a function of acquisition step for all methods. M-SA: MANATEE with scalarized acquisition, M-AS: MANATEE with acquisition of scalarized function, RA: random acquisition, RS: random scalarization. Solid line shows the mean and shaded region denotes the 95% confidence interval.

### A.3 Cross-validation experiments

In both experiments, subsampled and processed data (as described in Appendix B) was divided in 70/30% train/test splits in 5-fold cross-validation, where the parameter was optimized with MANATEE-SA only on train and cumulative regrets were computed in train and test. To ensure sufficient data, CITE-seq data was subsampled to 3000 cells in the cross-validation experiments. For the highly variable gene selection experiment, the same set of highly variable genes (computed according to the proportion value acquired on train at each step) was used when computing regrets on test. For both experiments, the initial dataset contained 5 training points at random locations and MANATEE-SA performed 10 acquisitions. Each cross-validation experiment was repeated 100 times. When computing regrets, the overall meta-objective maximum for ARI and NMI was set as the maximum of ARI/NMI values computed from the acquisitions by MANATEE-SA in each run and each fold. For each run, cumulative regret was averaged over folds. Supplementary Table 5 shows cross-validation cumulative regrets averaged over runs on train and test for both experiments.

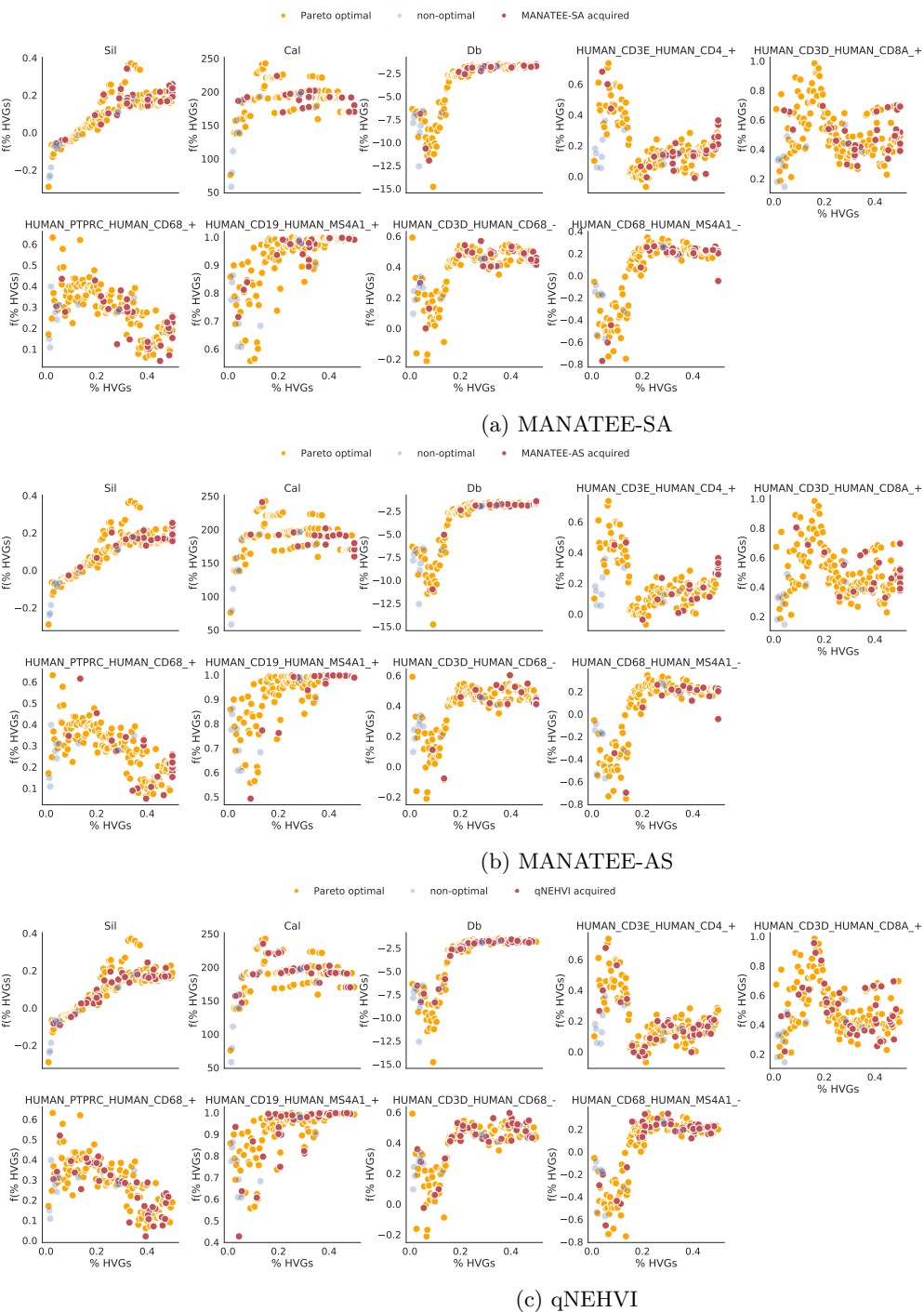

Supplementary Figure 8: 150 random samples from 9 objectives and meta-objectives (ARI, NMI) in the % HVGs selection experiment. (cont.)

## A.4 Behaviour ablation experiments

In leave-one-out behaviour experiments, MANATEE-SA was run without one of each behaviours at the time. Supplementary Table 6 shows results for the IMC cofactor selection experiment, with each row indicating

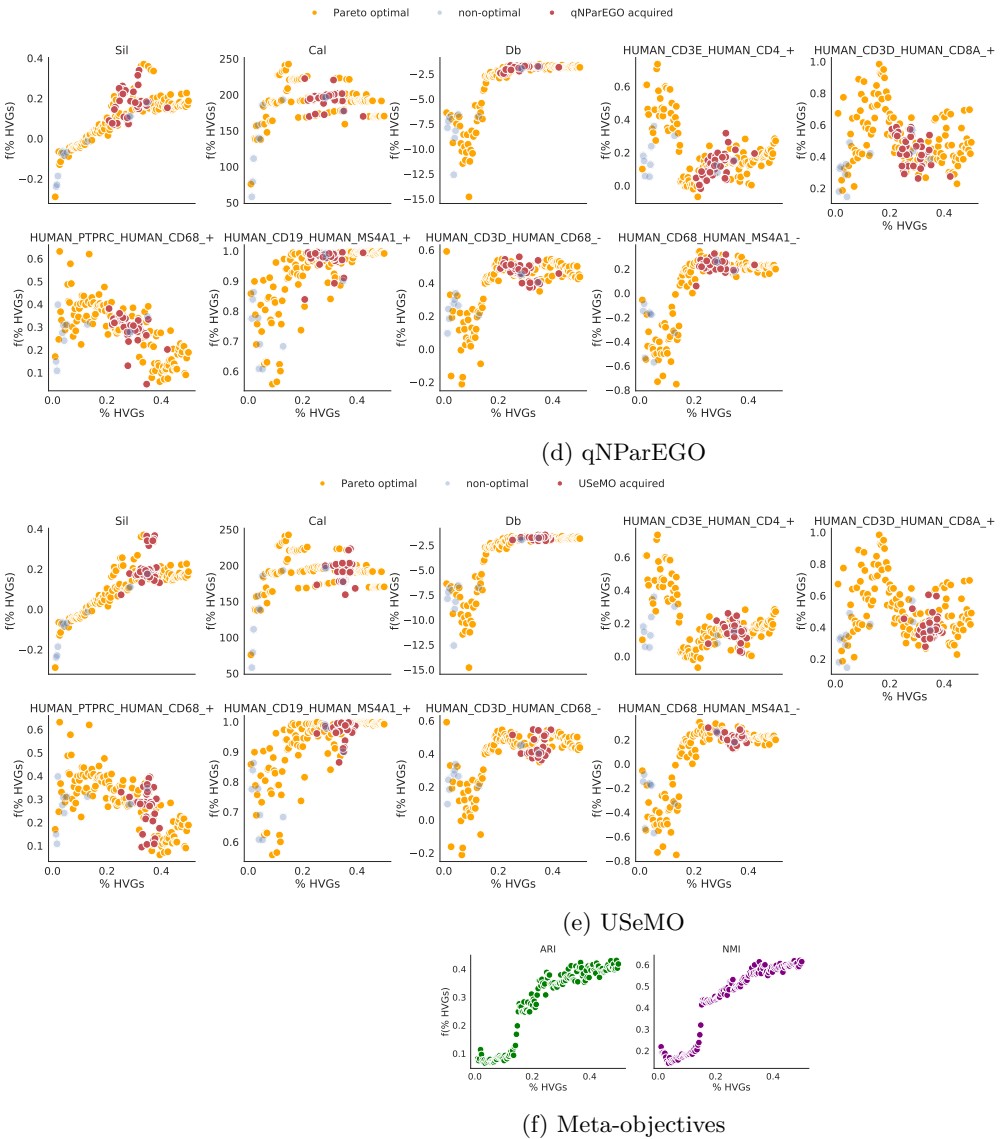

Supplementary Figure 8: 150 random samples from 9 objectives and meta-objectives (ARI, NMI) in the % HVGs selection experiment. Orange indicates samples on the Pareto front, blue indicates Pareto dominated samples, and red indicates points acquired by each method. The acquisitions shown for each method are from the run with the highest average ARI value. Pareto optimality of random samples was computed with the OApackage (Eendebak & Vazquez, 2019).

regrets for MANATEE-SA without said behaviour. Supplementary Table 7 similarly shows results for the scRNA-seq highly variable gene selection experiment.

## A.5 Computing regret curves

Regret curves for cumulative regret (CR), full regret (FR), and Bayes regret (BR) (denoted $\mathrm{CR}(t'), \mathrm{FR}(t'), \mathrm{BR}(t')$) were computed as a function of the acquisition step $t' = 1, \ldots, T$ as follows:

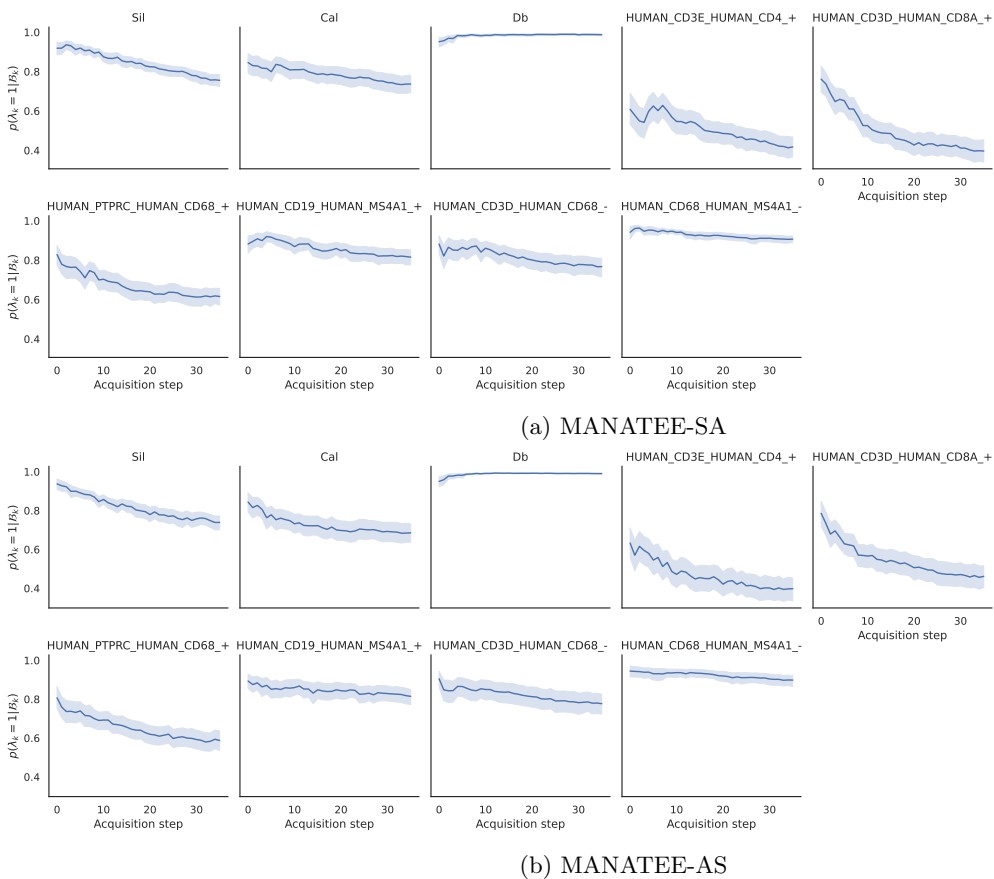

(a) MANATEE-SA

(b) MANATEE-AS

Supplementary Figure 9: Inclusion probabilities for MANATEE-SA and MANATEE-AS for each of the objectives as a function of acquisition step in the % HVGs selection experiment. Solid line shows the mean and shaded region denotes the 95% confidence interval across runs.

Supplementary Table 5: 5-fold cross-validation mean cumulative regret on train and test data splits. IMC denotes the IMC cofactor optimization experiment, scRNA-seq denotes the highly variable gene proportion optimization experiment. ARI: adjusted Rand index, NMI: normalized mutual information.

| | |
|---|---|
| IMC ARI train | $0.015 \pm 0.003$ |
| IMC ARI test | $0.017 \pm 0.003$ |
| IMC NMI train | $0.021 \pm 0.005$ |
| IMC NMI test | $0.023 \pm 0.005$ |
| scRNA-seq ARI train | $0.095 \pm 0.015$ |
| scRNA-seq ARI test | $0.089 \pm 0.015$ |
| scRNA-seq NMI train | $0.126 \pm 0.021$ |
| scRNA-seq NMI test | $0.126 \pm 0.020$ |

$$\mathrm{CR}(t') = \frac{1}{t'} \sum_{t=1}^{t'} (y^* - h(x_t)), \ \forall t' = 1, .., T$$

$$\mathrm{FR}(t') = y^* - \max_{x \in X_{1:t'}} h(x), \ \forall t' = 1, .., T$$

$$\mathrm{BR}(t') = \frac{1}{t'} \sum_{t=1}^{t'} (y^* - \max_{x \in X_{1:t'}} h(x)), \ \forall t' = 1, .., T$$

Supplementary Table 6: Results for the behaviour ablation experiments for IMC cofactor optimization. CR: cumulative regret, FR: full regret; BR: Bayes regret. ARI: adjusted Rand index, NMI: normalized mutual information. Values are mean (s.d.).

| Ablated | ARI | | | NMI | | |
|---|---|---|---|---|---|---|
| | CR | FR | BR | CR | FR | BR |
| Explainability | 0.016(0.004) | 0.002(0.003) | 0.007(0.004) | 0.017(0.006) | 0.002(0.005) | 0.007(0.007) |
| Inter-obj agreement | 0.018(0.005) | 0.003(0.005) | 0.007(0.006) | 0.020(0.009) | 0.003(0.008) | 0.008(0.010) |
| Max not at boundary | 0.018(0.007) | 0.004(0.006) | 0.008(0.007) | 0.020(0.012) | 0.004(0.011) | 0.009(0.012) |

Supplementary Table 7: Results for the behaviour ablation experiments for scRNA-seq highly variable gene selection. CR: cumulative regret, FR: full regret; BR: Bayes regret. ARI: adjusted Rand index, NMI: normalized mutual information. Values are mean (s.d.).

| Ablated | ARI | | | NMI | | |
|---|---|---|---|---|---|---|
| | CR | FR | BR | CR | FR | BR |
| Explainability | 0.126(0.023) | 0.055(0.012) | 0.063(0.012) | 0.121(0.032) | 0.021(0.009) | 0.030(0.012) |
| Inter-obj agreement | 0.129(0.018) | 0.055(0.011) | 0.063(0.012) | 0.126(0.025) | 0.020(0.010) | 0.030(0.014) |
| Max not at boundary | 0.120(0.022) | 0.055(0.012) | 0.063(0.013) | 0.113(0.033) | 0.020(0.010) | 0.030(0.013) |

Note that these cumulative quantities up to step $t'$ for $t' = 1, \ldots, T$ are normalized by the number of acquisition steps so far, leading to most curves decreasing with acquisition step. However, these quantities are not guaranteed to decrease with acquisition step for all methods as they are computed w.r.t. the meta-objectives (ARI, NMI), while all methods make acquisitions by having access only to heuristic objectives, which do not necessarily agree with the meta-objectives in our setup. The heuristic objectives and meta-objectives are also noisy functions owing to being computed using real noisy genomics data.

Regret curves for all methods for the IMC cofactor selection experiment are shown in Supplementary Figure 10a and those for the % HVGs selection experiment are shown in Supplementary Figure 10b.

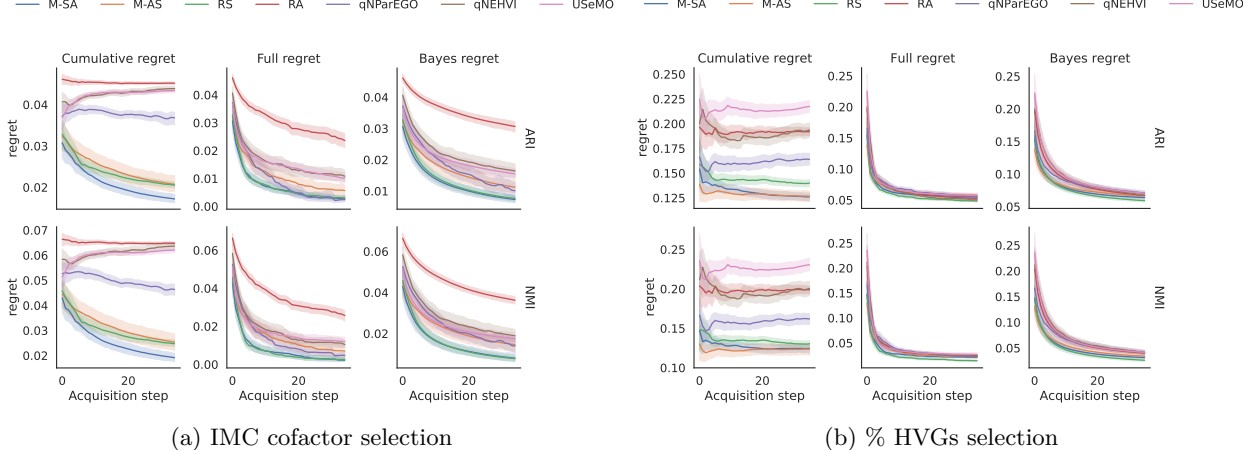

(a) IMC cofactor selection                    (b) % HVGs selection

Supplementary Figure 10: Normalized cumulative CR, BR, FR for ARI and NMI meta-objectives for all methods in both experiments. Solid line shows the mean and shaded region denotes the 95% confidence interval. CR: cumulative regret, FR: full regret; BR: Bayes regret. M-SA: MANATEE with scalarized acquisition, M-AS: MANATEE with acquisition of scalarized function, RA: random acquisition, RS: random scalarization. ARI: adjusted Rand index, NMI: normalized mutual information.

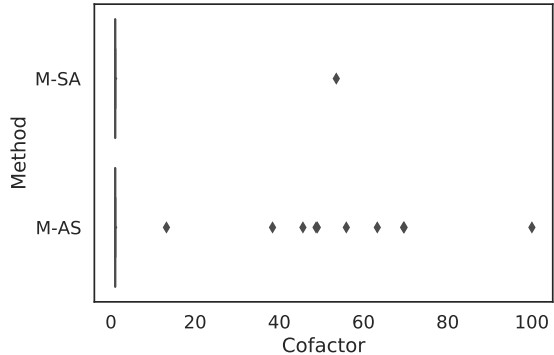

(a) Cofactor values selected in runs of the IMC cofactor selection experiment

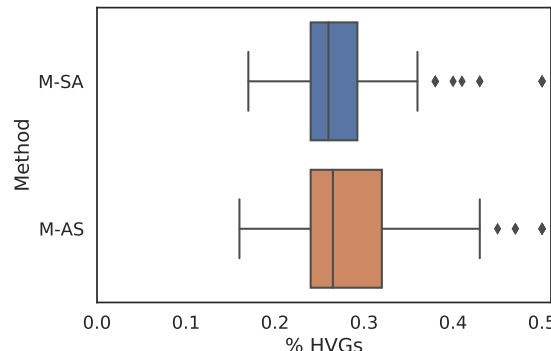

(b) Percentage values selected in runs of the % HVGs selection experiment

Supplementary Figure 11: Boxplots showing modes of the acquisitions made in each run by two MANATEE variants for a) IMC cofactor selection experiment and b) % HVGs selection experiment. The mode of the acquisitions made in a run represents the parameter value selected by a method in that run. Modes were computed on the acquisition values rounded to 2 decimal places. Each run (repetition) of an experiment had a different random seed controlling the initial training set and initialization of the Gaussian process model. M-SA: MANATEE-SA, M-AS: MANATEE-AS.

## A.6 Computing best achieved hypervolume

For each experiment, the hypervolume was computed with the `compute_hypervolume()` function of the `DominatedPartitioning` BoTorch class. At each acquisition step, we computed the dominated hypervolume of the dataset acquired so far w.r.t. the reference point. The acquired dataset included the initial points and the acquisitions made up to the current step. All acquisitions were scaled to $[0, 1]$ to account for different ranges of the objectives. To scale the Calinski and Harabasz score and the Davies-Bouldin score, we computed their upper and lower bounds (note that we define our objective as the negative Davies-Bouldin score) as the maximum and minimum acquired value across all methods in all runs, respectively. We defined the lower bound of the Calinski and Harabasz score and the upper bound of the negative Davies-Bouldin score as 0. For all correlation-based objectives and the silhouette width, we defined the lower and upper bounds as -1 and 1, respectively. For each experiment, we defined the reference point as the lower bound of each objective. Specifically, for the correlation-based objectives and the silhouette width the lower bound was -1, for the Calinski and Harabasz score the lower bound was 0, and for the Davies-Bouldin score the lower bound was set as the minimum of all acquisitions.

The curves showing the best achieved hypervolume at each iteration for all methods in both experiments are shown in Supplementary Figure 12. The best achieved hypervolume at each iteration was computed as the maximum hypervolume at that iteration across all runs. For the cofactor selection experiment, the maxima were computed across 98 runs for MANATEE-SA, RS, RA, USeMO; 96 runs for MANATEE-AS; 49 successfully terminated runs for $q$NEHVI; and 17 successfully terminated runs for $q$NParEGO. For the highly variable gene selection experiment, the maxima were computed across 100 runs for RA, USeMO; 99 runs for MANATEE-SA, MANATEE-AS, RS; 36 successfully terminated runs for $q$NEHVI; and 37 successfully terminated runs for $q$NParEGO.

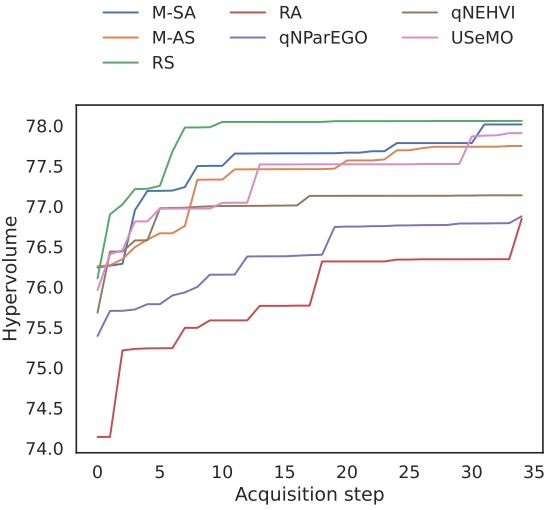

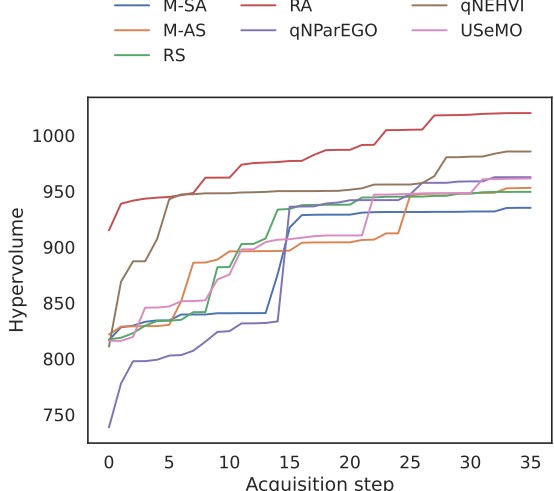

(a) Best achieved hypervolume in the IMC cofactor selection experiment

(b) Best achieved hypervolume in the % HVGs selection experiment

Supplementary Figure 12: Best achieved hypervolume at each acquisition step for a) IMC cofactor selection experiment and b) % HVGs selection experiment. Dominated hypervolume of the acquired dataset so far was computed for each acquisition step and the maximum was selected across all runs. Each run (repetition) of an experiment had a different random seed controlling the initial training set and initialization of the Gaussian process model. M-SA: MANATEE-SA, M-AS: MANATEE-AS, RA: random acquisition, RS: random scalarization.

# B Data processing

## B.1 Toy data

The 5 objectives in the toy experiment had the following functional forms all defined on $x \in [0, 1]$:

1. $y_1(x) = \max(0, \sin 2\pi x) + \epsilon, \ \epsilon \sim \mathcal{N}(0, 0.3^2)$

2. $y_2(x) = \sin 2\pi(x - 0.05) + \epsilon, \ \epsilon \sim \mathcal{N}(0, 0.3^2)$

3. $y_3(x) = \sin 2\pi x + \epsilon, \ \epsilon \sim \mathcal{N}(0, 0.3^2)$

4. $y_4(x) = -2x + \epsilon, \ \epsilon \sim \mathcal{N}(0, 0.8^2)$

5. $y_5(x) = 2x + \epsilon, \ \epsilon \sim \mathcal{N}(0, 0.1^2)$

## B.2 IMC data

IMC data and expert annotated ground truth clustering used in the experiments come from Jackson et al. (2020). Data was randomly subsampled to 5000 cells and the heavy metal markers that weren't conjugated to antibodies were removed. Data was clustered with the `scikit-learn` (Pedregosa et al., 2011) k-means algorithm with $k = 10$.

## B.3 CITE-seq data

CITE-seq data used in these experiments come from Stoeckius et al. (2017) retrieved using the `SingleCellMultiModal` Bioconductor R package with data version 1.0.0. Cell surface antibody expression was normalized using the `logNormCounts` from the `scuttle` R package (McCarthy et al., 2017) and

clustered using Seurat v. 4.1.0 (Hao et al., 2021) with top 10 principal components as input and resolution parameter set to 0.8. Intra-cellular single-cell RNA-seq data was filtered for genes with at least 100 reads and further processed with `scanpy` (Wolf et al., 2018). Data was randomly subsampled to 1000 cells (except for cross-validation experiment, where it was subsampled to 3000 cells) and normalized using `pp.normalize_total` with `target_sum=1e4`, `pp.log1p`, and `pp.scale`. `scanpy` was used to select highly variable genes, compute the neighbourhood graph with 10 neighbours and top 40 principal components, compute the PCA decomposition with default arguments, and compute Leiden clustering with resolution parameter set to 0.8.

## C  Derivations of acquisition functions

### C.1  Expectation of scalarization of the single-objective acquisition function of objectives (SA)

We define the SA acquisition function as $\mathbb{E}_{p(\boldsymbol{\lambda}|\mathcal{B})}\left[s_{\boldsymbol{\lambda}}(\mathrm{acq}_{\mathrm{UCB}}(\mathbf{f}(x)))\right]$ and derive the following expression:

$$
\begin{aligned}
\mathbb{E}_{p(\boldsymbol{\lambda}|\mathcal{B})}\left[s_{\boldsymbol{\lambda}}(\boldsymbol{\mu}(x) + \sqrt{\beta}\boldsymbol{\sigma}(x))\right] &= \mathbb{E}_{p(\boldsymbol{\lambda}|\mathcal{B})}\left[\sum_{k=1}^{K}\lambda_k(\mu_k(x) + \sqrt{\beta}\sigma_k(x))\right] \\
&= \sum_{k=1}^{K}\mathbb{E}_{p(\boldsymbol{\lambda}|\mathcal{B})}\left[\lambda_k\right](\mu_k(x) + \sqrt{\beta}\sigma_k(x)) \\
&= \sum_{k=1}^{K}p(\lambda_k = 1|\mathbf{B}_k)(\mu_k(x) + \sqrt{\beta}\sigma_k(x))
\end{aligned}
$$

For the SA acquisition function, the optimized expression simplifies to the sum over $K$ objectives of each objective's $\mathrm{acq}_{\mathrm{UCB}}(f_k(x))$ function value weighted by the probability of that objective being useful conditioned on its behaviours $p(\lambda_k = 1|\mathbf{B}_k)$. While the SA formulation only takes into account the posterior variance of each $f_k$, $k = 1,\ldots,K$, fitting a multi-output Gaussian process to data is still desirable as it ensures that the posterior form of $\mathbf{f}$ reflects our assumptions about the heuristic objectives, namely, that they should be maximized by similar values of $x$.

### C.2  Expectation of single-objective acquisition function of the scalarized objectives (AS)

We define the AS acquisition function as $\mathbb{E}_{p(\boldsymbol{\lambda}|\mathcal{B})}\left[\mathrm{acq}_{\mathrm{UCB}}(s_{\boldsymbol{\lambda}}(\mathbf{f}(x)))\right]$ and wish to derive the following expression:

$$
\mathbb{E}_{p(\boldsymbol{\lambda}|\mathcal{B})}\left[\mathrm{acq}_{\mathrm{UCB}}\left(\sum_{k=1}^{K}\lambda_k f_k(x)\right)\right]
$$

First, notice that,

$$
(\lambda_1 f_1(x), \ldots \lambda_K f_K(x)) \sim \mathcal{N}\left(\boldsymbol{\lambda}\boldsymbol{\mu}(x), \boldsymbol{\lambda}^T\boldsymbol{\lambda}\Sigma(x)\right),
$$

where $\boldsymbol{\mu}(x)$ is the posterior mean and $\Sigma(x)$ is the posterior covariance of $\mathbf{f}$ evaluated at some $x$.

Then, their sum is distributed as:

$$
\sum_{k}\lambda_k f_k(x) \sim \mathcal{N}\left(\sum_{k}\lambda_k\mu_k(x), \sum_{k}\lambda_k^2\Sigma_{kk}(x) + 2\sum_{1\leq i<j\leq K}\lambda_i\lambda_j\Sigma_{ij}(x)\right)
$$

Now, we use this to derive:

$$\mathbb{E}_{p(\boldsymbol{\lambda}|\mathcal{B})}\left[\mathrm{acq}_{\mathrm{UCB}}\left(\sum_{k=1}^{K}\lambda_k f_k(x)\right)\right]$$

$$= \mathbb{E}_{p(\boldsymbol{\lambda}|\mathcal{B})}\left[\sum_k \lambda_k \mu_k(x) + \sqrt{\beta}\cdot\sqrt{\sum_k \lambda_k^2 \Sigma_{kk}(x) + 2\sum_{1\leq i<j\leq K}\lambda_i\lambda_j\Sigma_{ij}(x)}\right]$$

$$= \sum_k p(\lambda_k = 1|\mathbf{B}_k)\mu_k(x) + \sqrt{\beta}\cdot\mathbb{E}_{p(\boldsymbol{\lambda}|\mathcal{B})}\left[\sqrt{\sum_k \lambda_k^2 \Sigma_{kk}(x) + 2\sum_{1\leq i<j\leq K}\lambda_i\lambda_j\Sigma_{ij}(x)}\right]$$

For the AS acquisition function, the UCB is applied to the weighted sum of $f_k$, $k = 1,\ldots,K$. The optimized expression contains the sum of the posterior means weighted by $p(\lambda_k = 1|\mathbf{B}_k)$ as before, but the variance term takes into account the posterior covariance of $\mathbf{f}$.

The proposed acquisition functions can differ e.g. within regions of search space, where those objectives that have negative posterior covariance with others but still have a large weight due to other behaviours, have large posterior variance. In these cases, the SA formulation will assign a larger value to those regions than the AS formulation.

### C.2.1 Computing the variance term

We compute the expectation of the variance term w.r.t. $p(\boldsymbol{\lambda}|\mathcal{B})$ by exhaustively considering all $K$-dimensional binary vectors $\boldsymbol{\lambda}^r$, $r = 1,\ldots,2^K$, which is feasible in our experiments with $K = 7$ and $K = 9$ objectives. Specifically, we evaluate the expectation as:

$$\mathbb{E}_{p(\boldsymbol{\lambda}|\mathcal{B})}\left[\sqrt{\sum_k \lambda_k^2 \Sigma_{kk}(x) + 2\sum_{1\leq i<j\leq K}\lambda_i\lambda_j\Sigma_{ij}(x)}\right]$$

$$= \sum_{\boldsymbol{\lambda}^r\in\{0,1\}^K} p(\boldsymbol{\lambda}^r|\mathcal{B})\sqrt{\sum_k (\lambda_k^r)^2 \Sigma_{kk}(x) + 2\sum_{1\leq i<j\leq K}\lambda_i^r\lambda_j^r\Sigma_{ij}(x)}$$

$$= \sum_{\boldsymbol{\lambda}^r\in\{0,1\}^K}\left(\prod_k p(\lambda_k = \lambda_k^r|\mathbf{B}_k)\right)\sqrt{\sum_k (\lambda_k^r)^2 \Sigma_{kk}(x) + 2\sum_{1\leq i<j\leq K}\lambda_i^r\lambda_j^r\Sigma_{ij}(x)}$$

In problems with large $K$ where this approach becomes infeasible, the expectation can be approximated with $S$ Monte Carlo samples of $\lambda_k^s \sim p(\lambda_k|\mathbf{B}_k)\ \forall\ k$ and $s = 1,\ldots,S$ as follows:

$$\mathbb{E}_{p(\boldsymbol{\lambda}|\mathcal{B})}\left[\sqrt{\sum_k \lambda_k^2 \Sigma_{kk}(x) + 2\sum_{1\leq i<j\leq K}\lambda_i\lambda_j\Sigma_{ij}(x)}\right]$$

$$\approx \frac{1}{S}\sum_{s=1}^{S}\sqrt{\sum_k (\lambda_k^s)^2 \Sigma_{kk}(x) + 2\sum_{1\leq i<j\leq K}\lambda_i^s\lambda_j^s\Sigma_{ij}(x)}$$

## D  Derivatives of a multi-output Gaussian process

The posterior mean of a single-output Gaussian process with $N$ noisy observations $\mathbf{y}$ at a new location $x_*$ is given by (Williams & Rasmussen, 2006):

$$\bar{f}_* = \mathbf{K}(x_*, X)\left(\mathbf{K}(X, X) + \sigma_\epsilon^2 I_N\right)^{-1}\mathbf{y}.$$

### D.1  First derivative

For the exponentiated quadratic kernel function, the first derivative of the posterior mean (which is also the mean of the distribution over derivatives of the GP posterior functions) is (McHutchon, 2013):

$$\frac{\delta \bar{f}_*}{\delta x_*} = \frac{\delta \mathbf{K}(x_*, X)}{\delta x_*} \boldsymbol{\alpha} = -\Lambda^{-1} \tilde{X}^T \left( \mathbf{K}(x_*, X)^T \odot \boldsymbol{\alpha} \right)$$

$$\boldsymbol{\alpha} = \left( \mathbf{K}(X, X) + \sigma_\epsilon^2 I_N \right)^{-1} \mathbf{y}$$

$$\tilde{X} = [x_* - x_1, \ldots, x_* - x_N]^T.$$

For $D$-dimensional input $x$, diagonal matrix $\Lambda$ collects lengthscales $l_d^2$, $d = 1 \ldots D$ on the diagonal, $\tilde{X}$ is an $N \times D$ matrix and $\odot$ represents element-wise multiplication. The resulting derivative $\frac{\delta \bar{f}_*}{\delta x_*}$ is $D$-dimensional, with each element corresponding to the $d$-th input dimension.

### D.2  Second derivative

The second derivative of the posterior mean is given by

$$\frac{\delta^2 \bar{f}_*}{\delta (x_*)^2} = \frac{\delta}{\delta x_*} \frac{\delta \mathbf{K}(x_*, X)}{\delta x_*} \boldsymbol{\alpha}$$

The second derivative of $k(x_*, x_i)$ is a $D \times D$ matrix given by (McHutchon, 2013):

$$\frac{\delta^2 k(x_*, x_i)}{\delta (x_*)^2} = \left( -\Lambda^{-1} + \Lambda^{-1} (x_* - x_i)(x_* - x_i)^T \Lambda^{-1} \right) k(x_*, x_i)$$

We stack these to compute a $D \times D \times N$ matrix $\frac{\delta^2 \mathbf{K}(x_*, X)}{\delta (x_*)^2}$ and multiply it by $\boldsymbol{\alpha}$ to compute the second derivative of the posterior mean. The resulting derivative $\frac{\delta^2 \bar{f}_*}{\delta (x_*)^2}$ is a $D \times D$ matrix with $(i, j)$-th element corresponding to $\frac{\delta^2 \bar{f}_*}{\delta (x_*)_i \delta (x_*)_j}$ (the second derivative w.r.t. dimensions $i$ and $j$ of $x_*$).

### D.3  One-dimensional input case

For one-dimensional input and lengthscale $l$, the second derivative of the posterior mean w.r.t. $x_*$ simplifies to:

$$\frac{\delta^2 \bar{f}_*}{\delta x_*^2} = \left( -\frac{1}{l^2} \mathbf{K}(x_*, X) + \frac{1}{l^4} \tilde{X}^T \odot \tilde{X}^T \odot \mathbf{K}(x_*, X) \right) \boldsymbol{\alpha}$$

### D.4  Multi-output Gaussian process

For a multi-output Gaussian process with $M$ objectives, we arrange observations as $MN \times 1$ array: $\mathbf{y} = [y_{11}, \ldots, y_{N1} \ldots y_{1M}, \ldots, y_{NM}]^T$. We also augment the auxiliary matrix $\tilde{X}$ to become an $MN \times D$ matrix: $\tilde{X} = [x_* - x_1, \ldots, x_* - x_N \ldots x_* - x_1, \ldots, x_* - x_N]^T$.

The multi-output kernel is defined as $\mathbf{K}^{\text{multi}} = \mathbf{K}^{\text{IO}} \otimes \mathbf{K}$ and the additive noise term is $\boldsymbol{D} \otimes I_N$, where $\boldsymbol{D}$ is the diagonal matrix with task-specific observation noises.

The first derivative of the posterior mean for a multi-output Gaussian process is $D$-dimensional for each of the $M$ tasks. The second derivative returns a $D \times D$ matrix for each of the $M$ tasks. In the computation of the second derivative, the last two dimensions of $\frac{\delta^2 \mathbf{K}^{\text{multi}}(x_*, X)}{\delta (x_*)^2}$ are flattened similarly to $\mathbf{y}$ before multiplication by $\boldsymbol{\alpha}$.

We note that the desired gradients can also be accessed via the automatic differentiation engine instead of using the analytically derived expressions as we have done here.

# E    Proof of theorem

**Theorem 3.1:** If $\mathbb{E}_{p(\lambda_k|\mathbf{B}_k)}[\lambda_k] > 0 \; \forall k$, then $x^* = \arg\max_x \mathbb{E}_{p(\boldsymbol{\lambda}|\mathcal{B})}[s_{\boldsymbol{\lambda}}(\mathbf{f}(x))]$ lies on the Pareto front of $\mathbf{f}$.

**Proof:** Let $\lambda_k \in \{0,1\}$ be a binary variable for $k = 1, \ldots, K$ with $\lambda_k \sim p(\lambda_k|\mathbf{B}_k)$ given by a Bernoulli distribution parameterized by a function of $\mathbf{B}_k$. Let us denote its expectation as $\psi_k := \mathbb{E}_{p(\lambda_k|\mathbf{B}_k)}[\lambda_k] > 0 \; \forall k$ and denote $\boldsymbol{\lambda} = [\lambda_1, \ldots, \lambda_K]$. We further assume $\lambda_k$ is independent from $\lambda_{k'}$ conditioned on its behaviours $\mathbf{B}_k \; \forall k, k' : k \neq k'$. We also denote the set of all behaviours $\mathcal{B} = \{\mathbf{B}_k\}_{k=1}^K$. Then, for a linear scalarization function $s_{\boldsymbol{\lambda}}$ and by linearity of expectation:

$$
\begin{aligned}
\mathbb{E}_{p(\boldsymbol{\lambda}|\mathcal{B})}[s_{\boldsymbol{\lambda}}(\mathbf{f}(x))] &= \mathbb{E}_{p(\boldsymbol{\lambda}|\mathcal{B})}\left[\sum_k \lambda_k f_k(x)\right] \\
&= \sum_k f_k(x) \cdot \mathbb{E}_{p(\boldsymbol{\lambda}|\mathcal{B})}[\lambda_k] \\
&= \sum_k f_k(x) \cdot \mathbb{E}_{p(\lambda_1,\ldots,\lambda_K|\mathbf{B}_1,\ldots,\mathbf{B}_K)}[\lambda_k] \\
&= \sum_k f_k(x) \cdot \left(\sum_{\lambda_k} \lambda_k \sum_{\lambda_i : i \in \{1,\ldots,K\}\setminus k} p(\lambda_1,\ldots,\lambda_K|\mathbf{B}_1,\ldots,\mathbf{B}_K)\right) \\
&= \sum_k f_k(x) \cdot \left(\sum_{\lambda_k \in \{0,1\}} \lambda_k \; p(\lambda_k|\mathbf{B}_k)\right) \\
&= \sum_k f_k(x) \cdot (1 \cdot p(\lambda_k = 1|\mathbf{B}_k) + 0 \cdot p(\lambda_k = 0|\mathbf{B}_k)) \\
&= \sum_k f_k(x) p(\lambda_k = 1|\mathbf{B}_k) \\
&= \sum_k f_k(x) \psi_k
\end{aligned}
\tag{8}
$$

As $\psi_k > 0 \; \forall k$, $\sum_k f_k(x)\psi_k$ is monotonically increasing in all $f_k(x)$. If any scalarization function $s_{\boldsymbol{\lambda}}$ is monotonically increasing in all coordinates $f_k(x)$, $x^* = \arg\max_x s_{\boldsymbol{\lambda}}(\mathbf{f}(x))$ lies on the Pareto front of $\mathbf{f}$ for $\mathbf{f} = \{f_k\}_{k=1}^K$ (Roijers et al., 2013; Zintgraf et al., 2015; Paria et al., 2019). Thus, the solution to $\max_x \mathbb{E}_{p(\boldsymbol{\lambda}|\mathcal{B})}[s_{\boldsymbol{\lambda}}(\mathbf{f}(x))] = \max_x \sum_k f_k(x)\psi_k$ lies on the Pareto front of $\mathbf{f}$. In other words, maximizing expected $s_{\boldsymbol{\lambda}}(\mathbf{f}(x))$ under $p(\boldsymbol{\lambda}|\mathcal{B})$ with our construction returns Pareto optimal solutions.

# F    Cluster mean co-expression as a heuristic

## F.1    Overview

The cluster mean co-expression heuristic is demonstrated in Figure 13. After clustering the single-cell data, we can consider the expression of two proteins which should be *markers* for a given cell type, i.e. they should either both be co-expressed or not expressed. An example of this is shown in Figure 13 (right). Consequently, the correlation in the cluster means is high. Conversely, if the clustering does not capture the cell types well, the correlation in the cluster means will be low (Figure 13 left). The opposite logic applies if two proteins should be mutually exclusively expressed: the correlation of cluster mean expression should be minimized.

## F.2    IMC experiment

The protein pairs used to construct co-expression heuristic objectives are listed in Supplementary Table 8.

## F.3    scRNA-seq experiment

The gene pairs used to construct co-expression heuristic objectives are listed in Supplementary Table 9.

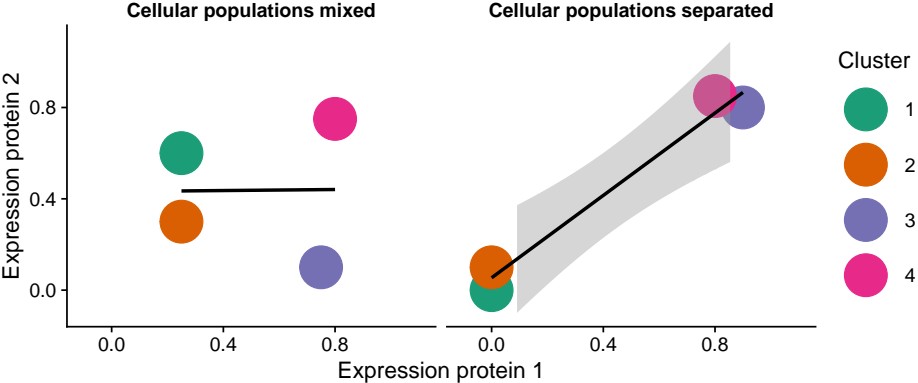

Supplementary Figure 13: Mean expression of two example proteins after two sets of clustering. Left: the cluster means poorly separate into double-positive and double-negative populations as would be expected if the two proteins are markers for the same cell type. Left: the ideal situation, where clusters only co-express both proteins simultaneously or not at all.

Supplementary Table 8: Co-expression of marker pairs used as objectives in the IMC cofactor selection experiment.

| Protein pair | Co-expression direction | Cell type |
|---|:---:|---|
| E-Cadherin, pan-Cytokeratin | + | Epithelial |
| CD45, CD20 | + | B cell |
| CD45, CD68 | + | Myeloid |
| CD45, CD3 | + | T cell |
| Vimentin, Fibronectin | + | Stromal cell |
| CD19, CD20 | + | B cell |
| pan-Cytokeratin, Cytokeratin 5 | + | Basal epithelial |

Supplementary Table 9: Co-expression of marker pairs used as objectives in the scRNA-seq highly variable gene selection experiment.

| Gene pair | Co-expression direction | Cell type(s) |
|---|:---:|---|
| CD3E, CD4 | + | Regulatory T cell |
| CD3D, CD8A | + | Cytotoxic T cell |
| PTPRC, CD68 | + | Myeloid |
| CD19, MS4A1 | + | B cell |
| CD3D, CD68 | - | T/myeloid |
| CD68, MS4A1 | - | Myeloid/B |

