# OpenReview forum: "Multi-objective Bayesian Optimization with Heuristic Objectives for Biomedical and Molecular Data Analysis Workflows"
_TMLR — Accepted by TMLR_

### Review · Reviewer_QD2q · 2023-01-08

**Summary Of Contributions:**

I have reviewed the previous version of this submission.

The current paper has addressed the outstanding concerns from last reviewing round satisfactorily. I recommend accepting the paper.

**Audience:**

Yes

**Claims And Evidence:**

Yes

**Requested Changes:**

N/A

**Strengths And Weaknesses:**

N/A

---

> ### Author Response · Authors · 2023-02-24
> **Reply to Review of Paper693 by Reviewer QD2q**
>
> > I have reviewed the previous version of this submission. The current paper has addressed the outstanding concerns from last reviewing round satisfactorily. I recommend accepting the paper.
>
> We thank the reviewer for the thoughtful suggestions that improved our work and for this positive assessment.

---

### Review · Reviewer_ZVWs · 2023-01-20

**Summary Of Contributions:**

This paper proposes MANATEE, an acquisition function for multi-objective Bayesian optimization, that infers positive qualities of different objective functions in terms of practitioner preferences, rather than being solely reliant on the hypervolume or other scalarizations of the several objective functions. In the experiments here, MANATEE focuses in on points with low predictive noise (what they call explainability), correlation between objectives, and non-boundary points.

The authors perform experiments with synthetic data and two biology tasks – mass cytometry cofactor detection and single cell gene selection. The method performs reasonably well across the objectives they measure across both cumulative regret (Fig. 10) and the objectives they were looking for (Tables 1 and 2). The biology tasks seem well connected to real problems despite being one dimensional and are designed around the concept of a meta-objectives, which softly measure optimization goals.


**Audience:**

Yes

**Claims And Evidence:**

Yes

**Requested Changes:**

Requested changes:

(medium) Can you doublecheck the cumulative regret plots in Fig. 10b? I’m not entirely sure they are going to be leveling off, but should continue upwards, or at least asymptoting?

(minor) Fig 10a: label these as cumulative regret, full regret, bayes regret.

(minor) Fig 5,8: maybe only display the pareto front or show the other points with a lower alpha value

(minor) fig 7: these should probably be all on the same plot to maximize differences

(medium) I would also suggest that the authors plot best achived hypervolume across iterations for the tasks as that seems to be a common concern.

Appendix d: I still think that this can be all achieved with autograd but it’s really a minor point.


**Strengths And Weaknesses:**

Full disclosure, I have reviewed this paper before and find that most of both my requested changes and that of the other reviewers have been made. Good work overall.

*Strengths:*

- Maximizing for predictive confidence (high explainability) and not moving towards the boundary in multi-objective problems is certainly a good idea.

- Overall, I think I now better understand the subtle idea of a meta-objective that we’re slowly moving the optimization towards. This seems generally applicable across many areas of biological design campaigns, especially when our proxy tasks are defective in some way.

- Experimental tasks are realistic and seem grounded in biological motivation.

- Botorch code is provided in a common framework.

- Thank you for providing the algorithm box.

*Weaknesses:*

- Only single dimensional problems are considered in the experiments. All things considered this isn’t that bad of an issue.

- The regret curves are only somewhat demonstrating superiority over observations (e.g. over the course of a design campaign), which is probably just most reflective of the noisiness of the tasks.

- This is somewhat of a fundamental weakness that can probably be addressed in the discussion: Inter-objective agreement seems like it might limit the practitioner’s need to make tradeoffs between competing objectives in favor of jointly suboptimal but reasonable performing objective values.

---

> ### Author Response · Authors · 2023-02-24
> **Reply to Review of Paper693 by Reviewer ZVWs**
>
> > Full disclosure, I have reviewed this paper before and find that most of both my requested changes and that of the other reviewers have been made. Good work overall.
>
> We thank the reviewer for the positive assessment of the changes we introduced to the manuscript following the review process. We also thank the reviewer for the time spent reviewing our work and for the latest suggestions that we address below.
>
> > (medium) Can you doublecheck the cumulative regret plots in Fig. 10b? I’m not entirely sure they are going to be leveling off, but should continue upwards, or at least asymptoting?
>
> We have verified that the plots in Supplementary Figure 10b are correct. We note that the plotted normalized cumulative quantities for the three regret measures we consider (CR, BR, FR) are not guaranteed to decrease to 0 or some asymptote with the acquisition step for all considered methods. The reason is the regrets are computed on the meta-objectives (ARI, NMI), while all methods only have access to the heuristic objectives which (as per the setup) may not be related to the meta-objective. Further, the high variance may be explained by the fact that all objectives (meta or heuristic) are computed from the analysis of real noisy genomic data. We have added a comment about this in Supplementary section A.5.
>
> > (minor) Fig 10a: label these as cumulative regret, full regret, bayes regret.
>
> > (minor) Fig 5,8: maybe only display the pareto front or show the other points with a lower alpha value
>
> > (minor) fig 7: these should probably be all on the same plot to maximize differences
>
> We have introduced these changes (fixed titles in Fig 10a; displayed non-optimal points in lower transparency in Figs 5,8; plotted acquisitions on the same plot in Figs 4,7).
>
> > (medium) I would also suggest that the authors plot best achived hypervolume across iterations for the tasks as that seems to be a common concern.
>
> We computed the best achieved hypervolume on the heuristic objectives at each acquisition step for each method and plotted these across iterations in Supplementary Figure 12, as requested. The details are provided in Supplementary section A.6. We note that our method does not optimize all heuristic objectives simultaneously and thus is not expected to achieve greater hypervolume on the heuristic objectives than other methods that do.
>
> > Appendix d: I still think that this can be all achieved with autograd but it’s really a minor point.
>
> We have added a note in Appendix D acknowledging this.
>
> Please note that the new changes can be found in the uploaded revision.

---

> > ### Comment · Reviewer_ZVWs · 2023-03-16
> > **Thanks**
> >
> > Thank you for the requested changes and comment about cumulative regret. My question there was mostly that cumulative regret is often calculated with the "best" achieved point, hence why it asymptotes. However, I can see that in these experiments, it may not asymptote as what is "best" with respect to the optimized objectives may not be the "best" with respect to the meta-objectives.

---

> > > ### Author Response · Authors · 2023-03-16
> > > **Reply to Reviewer ZVWs**
> > >
> > > Thank you for the clarification and we really appreciate your time in reviewing our work.

---

### Comment · Action_Editors · 2023-02-27
**Starting the discussion phase**

As you probably saw, we have now entered the discussion phase even though we have just two submitted reviews.

The E-i-Cs have ruled that we can proceed like this even though one of the reviewers of the original version has been unresponsive, without the need to add a fresh reviewer.

Reviewers: it would be great if you can also explicitly check that the requests of the missing reviewer have been addressed and comment about that explicitly (like Reviewer ZVWs has already done in their review).

---

### Decision · Action_Editors · 2023-03-16

**Recommendation:** Accept as is

**Comment:**

This resubmission was only reviewed by two reviewers because we were unable to reach one of the original reviewers.

Both reviewers find that the resubmission addresses all reviewer comments raised for the original submission and is publishable.

**Audience:**

Both active reviewers agree that the paper will be of interest at least for the multi-objective Bayesian optimisation community.

**Claims And Evidence:**

Both active reviewers agree that the claims are supported by sufficient evidence.